

# Gas-to-particle partitioning of major biogenic oxidation products from monoterpenes and real plant emissions

Georgios I. Gkatzelis [1], Thorsten Hohaus [1], Ralf Tillmann[1], Iulia Gensch [1], Markus Müller [2,4], Philipp Eichler [2†], Kang-Ming Xu [3], Patrick Schlag [1††], Sebastian H. Schmitt [1], Zhujun Yu [1], Robert Wegener[1], Martin Kaminski[1], Rupert Holzinger [3], Armin Wisthaler [2,5], Astrid Kiendler-Scharr [1]

[1] Institute of Energy and Climate Research, IEK-8: Troposphere, Forschungszentrum Jülich GmbH, Jülich, Germany
[2] Institut für Ionenphysik und Angewandte Physik, Universität Innsbruck, Innsbruck, Austria
[3] Institute for Marine and Atmospheric research Utrecht, Princetonplein 5, 3584 CC, Utrecht, The Netherlands
[4] Ionicon Analytik GmbH, Innsbruck, Austria
5 Department of Chemistry, University of Oslo, Norway

[†] Now at: German Environment Agency, Dessau-Roßlau, Germany

[††] Now at: Institute of Physics, University of Sao Paulo, Sao Paulo, Brazil

Correspondence to: T. Hohaus (t.hohaus@fz-juelich.de)

**Abstract.**

Secondary organic aerosols (SOA) play a key role in climate change and air quality. Determining the fundamental parameters that distribute organic compounds between the phases is essential, as atmospheric lifetime and impacts change drastically between gas- and particle-phase. In this work, gas-to-particle partitioning of major biogenic oxidation products was investigated using three different aerosol chemical characterization techniques. The aerosol collection module (ACM), the collection thermal desorption unit (TD) and the chemical analysis of aerosol on-line (CHARON) are different aerosol sampling inlets connected to a Proton Transfer Reaction-Time-of-Flight-Mass Spectrometer (PTR-ToF-MS). These techniques were deployed at the atmosphere simulation chamber SAPHIR to perform experiments on the SOA formation and aging from different monoterpenes (β-pinene, limonene) and real plant emissions (*Pinus sylvestris L.*). The saturation mass concentration C* and thus the volatility of the individual ions was determined based on the simultaneous measurement of their signal in the gas- and particle-phase.

A method to identify and exclude ions affected by thermal dissociation during desorption and ionic dissociation in the ionization chamber of the PTR-MS was developed and tested for each technique. Narrow volatility distributions with organic compounds in the semi-volatile (SVOCs) to intermediate volatility (IVOCs) regime were found for all systems studied. Despite significant differences in the aerosol collection and desorption methods of the PTR based techniques, comparison of the C* values obtained with different techniques were found to be in good agreement (within 1 order of magnitude) with deviations explained by the different operating conditions of the PTRMS.



The C* of the identified organic compounds were mapped onto the 2-dimensional volatility basis set (2D-VBS) and results
showed a decrease of the C* with increasing oxidation state. For all experiments conducted in this study, identified
partitioning organic compounds accounted for 20-30 % of the total organic mass measured from an AMS. Further
comparison between observations and theoretical calculations was performed for species found in our experiments that were
also identified in previous publications. Theoretical calculations based on the molecular structure of the compounds showed,
within the uncertainties ranges, good agreement with the experimental C* for most SVOCs, while IVOCs deviated up to a
factor of 300. These latter differences are discussed in relation to two main processes affecting these systems: (i) possible
interferences by thermal and ionic fragmentation of higher molecular weight compounds, produced by accretion and
oligomerization reactions, that fragment in the *m/z* range detected by the PTRMS and (ii) kinetic influences in the
distribution between gas- and particle-phase with gas-phase condensation, diffusion in the particle-phase and irreversible
uptake.

## 1   Introduction

Secondary organic aerosol (SOA), formed by chemical reactions in the atmosphere, constitute a major fraction of the organic
aerosol ((OA); Jimenez et al 2009)) and thus play a key role in climate change and air quality. A detailed understanding of
SOA formation and composition needs to be well defined for impact mitigation (de Gouw et al., 2008; Hallquist et al., 2009;
Jimenez et al., 2009; Volkamer et al., 2006). Defining the fundamental parameters that distribute organic molecules between
the gas  and particle phases is essential, as atmospheric lifetime and impacts change drastically between phases. The
saturation vapor pressure (James F. Pankow, 1994) and the enthalpies of vaporization and sublimation are key
thermodynamic properties describing the gas-to-particle partitioning of organic compounds. Since SOA consists
predominantly of oxidized multifunctional compounds (McFiggans et al., 2010) they are expected to show low saturation
vapor pressures thus increasing the detection challenges due to the low gas-phase concentrations that need to be probed
(Bilde et al., 2015).
Advanced instrumentation for defining the saturation vapor pressure and thus the volatility of single component and complex
organic aerosol systems has been developed in the past decades both for laboratory and field studies. Dicarboxylic acids
represent a class of low-volatility compounds commonly found in atmospheric aerosol that are commercially available.
These molecules have been extensively studied by various techniques (Bilde et al., 2015). Namely, the Knudsen effusion
mass spectrometry (KEMS) (Booth et al., 2009) is a method where macroscopic crystalline samples effuse in a Knudsen cell
and the change of the concentration in the gas-phase is measured using a mass spectrometer and translated to saturation
vapor pressure values based on calibrated standards. Single particle methods using optical tweezers (Mitchem and Reid,
2008) and the electrodynamic balance (EDB) (Pope et al., 2010) infer saturation vapor pressure values from the evaporation
or condensational growth of a single particle at a controlled environment. Thermal desorption mass spectrometry (TDMS)
has extended the studies from laboratory to ambient complex polydisperse systems. Thermodenuders have been extensively



used to quantify the volatility of the bulk OA (An et al., 2007; Faulhaber et al., 2009; Gkatzelis et al., 2016; Huffman et al.,
2008; Isaacman-VanWertz et al., 2017; Louvaris et al., 2017) with the support of model calculations (Karnezi et al., 2014;
Riipinen et al., 2010). However, the detector used in most of these studies is an Aerosol Mass Spectrometer (AMS)
(Canagaratna et al., 2007) that operates at high vaporizer temperatures (600 °C) and ionizes the analytes by electron impact
(70 eV) thus introducing excessive thermal and ionic dissociation.
Recently, several different methods have been developed that compromise between molecular level information for a small
fraction of the OA mass (Hohaus et al., 2010; Kreisberg et al., 2009; Williams et al., 2006; Williams et al., 2014; Zhang et
al., 2014) or chemical formula identification using soft ionization MS to achieve a nearly full OA characterization (Gkatzelis
et al., 2017; Isaacman-VanWertz et al., 2017; Lopez-Hilfiker et al., 2014; Stark et al., 2017). Volatility measurements are
performed either by calibrating with standards of known saturation vapor pressure (Lopez-Hilfiker et al., 2015; Lopez-
Hilfiker et al., 2014) or by simultaneous measurement of the gas- and particle-phase of an ion when applicable (Hohaus et
al., 2015; Isaacman-VanWertz et al., 2016; Stark et al., 2017).
In order to identify the OA on a molecular level, thermal desorption techniques have been coupled to Gas-Chromatography
(GC) methods. The Thermal Desorption Aerosol Gas Chromatograph/Mass Spectrometer (2D-TAG) (Isaacman et al., 2011)
and the Volatility and Polarity Separator (VAPS) (Martinez et al., 2016) are similar techniques that provide volatility- and
polarity-resolved OA information by using a 2-dimensional gas chromatography (2D-GC) approach. Volatility is derived
based on the two-dimensional chromatographic retention times relative to those of know standards, thus establishing a
retention time correlation (RTC) to the vapor pressure. Simultaneous measurements of the gas- and particle-phase mass of
organic molecules has also been recently developed using the modified semi-volatile TAG (SV-TAG) that utilizes two TAG
cells in parallel (Isaacman-VanWertz et al., 2016). Although the above GC methods provide chemical speciation and gas-to-
particle partitioning in a molecular level, they can only do so for a small fraction of the OA mass (10 – 40 %).
Newly developed thermal desorption inlets have allowed near-simultaneous chemical characterization of gas- and particle-
phase ambient compounds (Eichler et al., 2015; Gkatzelis et al., 2017; Holzinger et al., 2010; Lopez-Hilfiker et al., 2014;
Stark et al., 2017; Yatavelli et al., 2014). When coupled to chemical ionization high resolution time-of-flight mass
spectrometers (ToF-CIMS) these inlets can provide information on a very broad volatility range (Eichler et al., 2017). By
simultaneous measurement of the gas- and particle-phase mass concentration when applicable, direct volatility calculations
of individual species can be performed. Indirect ways of estimating the vapor pressure for this type of systems has been also
established based on the desorption temperature of calibrated known species or mixtures (Lopez-Hilfiker et al., 2016; Stark
et al., 2017). Since the above mass spectrometric techniques can provide elemental formulas, methods to derive the vapor
pressure by assuming a functional group composition have also been widely used (Krechmer et al., 2015; J. F. Pankow and
Asher, 2008). A detailed comparison of the three different methods of estimating the vapor pressure for this type of
techniques has been performed for field studies under forested areas (Stark et al., 2017). Results suggested that thermal
decomposition pathways could bias the direct partitioning calculation based on the gas- and particle-phase concentrations as





well as calculations based on the chemical formula of the species detected. Detailed understanding on the decomposition pathways is to be determined in future studies.

There are two major ways established in the last years to treat partitioning for practical applications to atmospheric aerosol. One is through a thermodynamic model containing an ensemble of specific molecules while the other is based on empirical calculations (Donahue et al., 2014). When using explicit methods, model systems are treated as fully as possible thus individual vapor pressures and activity coefficients are calculated based on several thermodynamic schemes (Clegg et al., 2001; Fredenslund et al., 1975; Zuend et al., 2011). These calculations are strongly affected by the wide range of vapor pressure estimates from the different theoretical approaches (Camredon et al., 2010; Donahue et al., 2014), thus further promoting the need of future development in this field. On the contrary, empirical methods tend to simulate gas-to-particle partitioning based on fits of partitioning data derived from chamber observations. Frameworks like the 2-Dimensional Volatility Basis Set (2D-VBS) classify OA in terms of bulk chemical characteristics and volatility (Donahue et al., 2013; Donahue et al., 2012). A variety of the above newly developed techniques can be mapped onto the 2D-VBS and thus provide important experimental input to further develop and test both the empirical methods and the newly developed instrumentation.

Deviations of the theoretical to experimental vapor pressure estimates are systematically observed (Bilde et al., 2015). Recent measurements show enrichment of semi-volatile organic compounds in the particle- relative to the gas-phase than calculations based on equilibrium vapor pressure would suggest (Hohaus et al., 2015; Isaacman-VanWertz et al., 2016; Zhao et al., 2013). It is currently unclear whether this is due to (i) uncertainties in the theoretical estimates of vapor pressures, (ii) thermal decomposition pathways affecting the experimental partitioning determination or (iii) the existence of uptake pathways to particles other than absorption e.g. adsorption or reactive uptake. The wide range of theoretical vapor pressure estimates combined with the large gas-to-particle partitioning discrepancies of the above techniques (Thompson et al., 2017) promote further studies in order to bridge the gap between theory and experiments.

In this study, the gas-to-particle partitioning of major biogenic SOA (BSOA) oxidation products was investigated. An inter-comparison was performed using three different inlet techniques connected to soft-ionization mass spectrometry, the Aerosol Collection Module (ACM) (Hohaus et al., 2010), the Chemical Analysis of Aerosol Online (CHARON) (Eichler et al., 2015) and the Collection Thermal Desorption Cell (TD) (Holzinger et al., 2010). Volatility measurements were derived based on the mass concentration of individual species in the gas- and particle-phase, implemented in the 2D-VBS and compared to various explicit methods.

## 2 Methods and instrumentation

### 2.1 Facilities

The SAPHIR chamber is an atmospheric simulation chamber made of a double walled Teflon (FEP) foil with a volume of 270 m³. It has a cylindrical shape and is housed in a steel frame. A shutter system mounted on the steel frame allows to



conduct experiments in the dark or when opened exposes the chamber to natural sunlight to initiate photochemistry. The
pressure inside the chamber is kept at about 50 Pa overpressure compared to ambient pressures to ensure no diffusion from
trace gases from the outside into the chamber. Additionally, the interspace of the double walled Teflon film is continuously
flushed with pure nitrogen (Linde, purity 99.9999 %). A continuous flow of ultra clean air into the chamber compensates any
losses due to leakages and ensures stable pressure conditions.
In preparation for each experiment the chamber is flushed with a high flow of up to 250 m³/h for several hours using the
ultra-clean air. The same high flow rate is used to humidify the chamber before the start of each experiment. For
humidification Milli-Q water is boiled and the steam is added to the air stream into the chamber. Two fans mounted inside
the chamber generate well mixed starting conditions and were turned off as soon as aerosol production was initated to reduce
aerosol losses in the chamber. Ozone is added using a silent discharge ozonizer. Standard instrumentation is continuously
measuring the conditions inside the SAPHIR chamber. Instrumentation includes an ultrasonic anemometer (Metek USA-1,
accuracy 0.3 K) to measure the air temperature, a frost point hygrometer (General Eastern model Hygro M4) to determine
the humidity, and a chemiluminescence analyser (ECO PHYSICS TR480) equipped with a photolytic converter (ECO
PHYSICS PLC760) to measure NO and $NO_2$. Ozone is measured by an UV absorption spectrometer (ANSYCO model
O341M). Further details of the SAPHIR chamber are described in Rohrer et al. (2005).
Precursor compounds were added using two separate methods. The first method was to inject pure liquid compounds via a
syringe in a heated inlet line, into the air stream with which the vapors were transported into the chamber. The second
method was to use the plant chamber SAPHIR-PLUS (Hohaus et al., 2016) to transfer the emissions of six *Pinus sylvestris L.*
(scots pine) into the chamber. Injection flow from SAPHIR-PLUS was 6 m³/h which replaced to a large extend the flow of
clean air (8 m³/h) which is needed to replace air lost due to leakage and withdrawal of analytical instrumentation. The
environmental parameters of the plant chamber are fully controlled (e.g., temperature, soil relative humidity,
photosynthetically active radiation). The average temperature inside the SAPHIR-PLUS chamber was 25 °C. Details on the
SAPHIR-PLUS are provided in Hohaus et al. (2016).
**2.2 Instrumentation**
All instruments used in this study are described in detail in Gkatzelis et al. (2017) and only an overview is provided in the
following. An Aerodyne High-Resolution Aerosol Mass Spectrometer (HR-AMS) (Canagaratna et al., 2007; DeCarlo et al.,
2006) and a Scanning Mobility Particle Sizer (SMPS, TSI Classifier model 3080, TSI DMA 3081, TSI Water CPC 3786),
were used to determine the aerosol chemical composition  including the total organic mass concentration and the size
distribution during the experiments, respectively. In order to determine the saturation mass concentrations (C*) parallel gas-
and particle-phase measurements were performed. Particle-phase composition was measured using three different aerosol
sampling techniques coupled to a Proton-Transfer-Reaction Time-of-Flight Mass Spectrometer (model PTR-TOF 8000;
PTR-ToF-MS, Ionicon), the Aerosol Collection Module (ACM-PTR-ToF-MS, referred to as "ACM" hereafter) (Hohaus et
al., 2010), the chemical analysis of aerosol online (CHARON-PTR-ToF-MS, referred to as "CHARON" hereafter) (Eichler



et al., 2015) and the collection thermal desorption unit (TD-PTR-ToF-MS, referred to as "TD" hereafter) (Holzinger et al.,
2010). In the following, the most important characteristics and parameters are described briefly and for more details the
reader is referred to Gkatzelis et al. (2017). The CHARON is a real time measurement (10 s integration time in the detector),
while the ACM and TD have sampling times for this study of 120 min and 240 min, respectively. The CHARON inlet was
operated at low pressure (< 1 atm) and at a constant temperature of 140 °C. The sampling in the ACM was under vacuum
conditions and at sub-zero temperature (-5 °C). The sampling in the TD was at ambient temperature and at atmospheric
pressure. The CHARON used a gas-phase denuder to strip off gas-phase compounds while the AMS-type vacuum inlet
system of the ACM ensured a removal of the gas-phase. While the particle-phase in the CHARON was desorbed by passing
particles through a thermodenuder, the particle-phase in the ACM and TD was desorbed after collection from the collection
surface using a temperature ramp reaching a final temperature of 250 °C and 350 °C, respectively. All three aerosol
collection techniques are utilizing a PTR-ToF-MS as a detector. The operational conditions for each PTR-ToF-MS were
different with regard to a different electric field strength (V cm$^{-1}$) to buffer gas density (molecules cm$^{-3}$) ratio (E/N). The
PTR-ToF-MS of the CHARON, ACM, and TD were operated at 100 Td, 120 Td, 160 Td, respectively (1 Td = $10^{-17}$ V cm$^2$
molecule$^{-1}$). The drift tube conditions for the PTR-ToF-MS of CHARON, ACM, and TD were at a temperature of 100 °C
with a pressure of 2.30 mbar, 120 °C and a pressure of 2.40 mbar, and 120 °C and a pressure of 2.40 mbar, respectively. The
limit of detection (LOD), depended on the different pre-concentration factors for each technique, which resulted in TD
having the lowest LOD ($10^{-3}$ ng m$^{-3}$), followed by the CHARON (1.4 ng m$^{-3}$), while ACM showed the highest values
(250 ng m$^{-3}$).
Gas-phase organic compounds were detected by a standalone PTR-ToF-MS for the CHARON and TD. The standalone PTR-
ToF-MS was operated with an E/N = 120 Td. The drift tube was kept at a temperature of 60 °C and a pressure of 2.30 mbar.
The standalone PTR-MS was connected to SAPHIR via a 0.5 m PFA line (inner diameter, i.d. 3.2 mm), to a 2 m PEEK line
(i.d. 1 mm), heated at 60 °C with a flow of 0.6 L min$^{-1}$ that resulted to an overall residence time of ~ 0.6 s. The ACM was
connected via a 4 m PFA line (i.d. 4 mm), at room temperature with a flow of 0.7 L min$^{-1}$, resulting to a residence time of
approximately 3 s. A PTFE particle filter (Merck Millipore) was additionally introduced to the line to reassure complete
particle-phase removal. Gas-phase compounds were then directed to the ACM-PTR-MS interface that was heated at 280 °C
via a 5 cm coated stainless steel line (i.d. 0.8 mm) to the PTR-MS. The ACM design allowed for simultaneous gas-phase
measurements with the same PTR-ToF-MS while sampling of the particle-phase took place on the ACM collector.
Comparison of gas and particle-phase measurements was thus performed using the same detector avoiding any detector
related differences. It should be noted that TD and CHARON are also designed for simultaneous gas- and particle-phase
measurements using the same PTR-MS but in this study this feature was not operational.
**2.3 Experimental conditions**
Before each experiment the SAPHIR chamber was flushed with clean air over night (total volume exchange was about 2000
m³) and humidified directly after the flushing process. Relative humidity (RH) in the chamber was about 55 % within a



temperature range for all experiments between 295 K and 310 K. After ensuring that all instruments had measured the
background in the SAPHIR chamber a single monoterpene (β-pinene or limonene), a monoterpene mixture (β-pinene and
limonene) or the emissions of 6 *Pinus sylvestris L.* (Scots pine) were injected. The tree emissions were characterized using
GC-MS. The composition of the biogenic VOC (BVOC) consisted of 42% δ3-carene, 38% α-pinene, 5% β-pinene, 4%
myrcene, 3% terpinolene and 8% other monoterpenes. The details of all experiments are given in Table 1 and an
experimental overview is provided in Figure S1. One hour after injection of the VOCs ozone was introduced into the
chamber to initiate ozonolysis with the subsequent formation of secondary organic aerosols (SOA). Experiments were done
without the use of an OH scavenger. $NO_x$ concentrations during the experiments ranged between 10 to 60 pptV originitating
from HONO background source (Rohrer et al., 2005). In all experiments, except for the experiment with limonene as a
precursor, 20 hours after the start of the ozonolysis the roof of the SAPHIR chamber was opened to initiate additionally
photochemistry with OH and ageing of the SOA. For the limonene experiment instead of opening the roof, 30 ppbV of NO
was added to the chamber. With the remaining ozone in the chamber, $NO_3$ oxidation was initiated. In the tree emissions
experiment the SAPHIR-PLUS chamber was recoupled to the SAPHIR chamber 11 hours after the start of the ozonolysis
thus injecting again fresh BVOC emissions from the scots pines. The experiment continued for an additional 6 hours with the
roof open allowing for further oxidation of the BVOCs and SOA by OH radicals. The duration of the experiments varied
from 17 to 36 hours, providing ample time to experimentally investigate the aging of the biogenic SOA.

## 2.4 Estimation of volatility distribution

In this work the volatility of different species was quantified based on their saturation mass concentration C* in units of
μg m$^{-3}$. Theoretical calculation of the saturation concentration was performed for known oxidation products of the
monoterpenes studied based on their chemical structure as seen in Table S1. Based on the absorption equilibrium partitioning
formalism, the (sub-cooled liquid) saturation vapor pressure ($p_{i,L}$) of a species was related to its C* based on Cappa and
Jimenez (2010) as following:
$$C^*(T) = \frac{MW_i \times 10^6 \times p_{i,L} \times \zeta_i}{R \times T} \qquad (1)$$
where $MW_i$ is the molecular weight of a compound i (g mol$^{-1}$), $p_{i,L}$ is the sub-cooled liquid saturation vapor pressure (Pa), $\zeta_i$
is the activity coefficient of species i in the OA phase, T is the chamber temperature (K) and R is the ideal gas
constant (8.314 J mol$^{-1}$ K$^{-1}$). Here, the calculations were performed using a mean molecular weight of 180 g mol$^{-1}$ (Prisle et
al., 2010). In conformity with Donahue et al. (2014) the activity coefficients of all considered species partitioning into a
mixed aerosol system containing similar compounds were assumed to be 1 throughout the study.
At present, there is a scarcity of reliable saturation vapor pressure data obtained through laboratory studies (Bilde et al.,
2015). Therefore, $p_{i,L}$ is commonly estimated using empirical relationships derived from the Clausius-Clapeyron equation
e.g. (Jenkin, 2004; Myrdal and Yalkowsky, 1997; Nannoolal et al., 2008). The required thermodynamic properties, such as





the boiling temperature or the enthalpy of vaporization are predicted from the molecular structure of the investigated
compounds (Joback and Reid, 1987; Mackay et al., 1982; Stein and Brown, 1994). Their explicit manual calculation using
the existing functional group contribution methods are very laborious not only because of the high number of components,
but also because of the wide range of multifunctional organic compounds in the aerosol mixtures. Recently, a new web-
based facility, UManSysProp (http://umansysprop.seaes.manchester.ac.uk), was developed for automating predictions of i.e.
pure component vapor pressures of organic molecules or activity coefficients for mixed liquid systems. For the group
contribution approaches, only the molecular information must be uploaded in form of SMILES (Simplified Molecular Input
Line Entry System) strings (Toppings et al., 2016). At a defined temperature, there are several options for vapor pressure
predictive techniques, providing the possibility to combine two different empirical representations of the Clausius-Clapeyron
equation (Myrdal and Yalkowsky, 1997; Nannoolal et al., 2008) - further referred to as MY and NN, respectively - with
three different prediction methods for thermodynamic properties of the investigated compound based on their molecular
structure (Joback and Reid, 1987; Nannoolal et al., 2008; Stein and Brown, 1994) - further referred to as JR, NN and SB,
respectively. Additionally, the EVAPORATION method (further referred to as EVAP) proposed by Compernolle et al.
(2010) is available for the web-based calculations.
Here, we use the $p_{i,L}$ predicted online by UManSysProp facility, examining all seven estimation methods (Figure S6). Only
the results giving the lowest and highest vapor pressures for the studied compounds (the range of the estimates are indicated
by the grey background color) are used in this study to compare measurements to the highest and lowest possible theoretical
values.
Experimental determination of the saturation mass concentration of the individual compounds was derived by applying the
partitioning theory (James F. Pankow, 1994) based on Donahue et al. (2006) as
$$\mathbf{C}^* = \mathbf{OA} \times \frac{G_i}{P_i} \qquad (2)$$
where OA is the total organic mass (µg m$^{-3}$) determined from AMS and $G_i$ and $P_i$ are the gas- and particle-phase mass
concentration (µg m$^{-3}$) of compound i, respectively, measured from the PTR based techniques. Assuming typical
vaporization enthalpies (Epstein et al., 2010), C* and therefore partitioning is strongly dependent on temperature with
changes of ± 15 °C resulting in a C* change of ± 10 µg m$^{-3}$. During the campaign the average chamber temperatures and
their standard deviations were 20 ± 4 °C, 17 ± 4 °C, 19 ± 5 °C and 30 ± 5 °C for the β-pinene, limonene, mixture and trees
experiment, respectively. Nevertheless, these variations (< 10 °C) can be considered small. Therefore, for consistency with
other studies, a reference temperature of 298 K was used throughout all C* calculations. This was recently proposed by Stark
et al. (2017) to derive the average C* for the BEACHON and SOAS field campaigns making the assumption that deviations
due to temperature changes (18 ± 7 °C and 25 ± 3 °C, respectively) were within the uncertainties of the measurements.



## 3   Results and discussion

### 3.1   Compound selection: Assessment of ionic and thermal decomposition

For all PTR based techniques the molecular formula $(C_xH_yO_zN_a)H^+$ was attributed to each detected signal derived from the exact molecular mass which was determined by the TOF-MS. Whether the detected ion was an original SOA compound or a fragment detected on this mass could be affected by two major processes, (i) thermal dissociation during desorption, and (ii) ionic dissociation in the ionization region of the PTR-ToF-MS.

Thermal dissociation has been found to introduce a high degree of fragmentation for compounds that contain multiple functional groups, including peroxide groups which are thermally labile (Lopez-Hilfiker et al., 2015). For organic alcohols and acids thermal desorption has been shown to lead to loss of carboxyl- ($-CO_2$), carbonyl- ($-CO$) and water-groups ($-H_2O$) (Canagaratna et al., 2015). Accretion reactions and gas-phase autoxidation have been found to play a key role in extremely low volatility OC (ELVOC) formation (Ehn et al., 2014; Tobias and Ziemann, 1999; 2001). Upon heating, such products will thermally decompose (Barsanti et al., 2017) and be detected in the lower molecular weight range, thus directly affecting the partitioning estimation (Jang and Kamens, 2001; Stark et al., 2017). All instruments deployed in this study were subjected to possible thermal dissociation with decarboxylation and dehydration reactions strongly dependent on the temperature, pressure and the heat exposure time of the molecules during desorption. CHARON was operated at the lowest temperature of 140 °C, under a few mbars of pressure and with the lowest heat exposure time, therefore minimizing the latter reactions. ACM and TD were operated at 1 bar and up to 250 °C and 350 °C, respectively, with longer heat exposure times.

Functional group loss has been found to additionally occur in the ionization region of the PTR-ToF-MS instruments. Gkatzelis et al. (2017) showed that for this study the ratio of the electric field strength (V cm$^{-1}$) to buffer gas density (molecules cm$^{-3}$) (E/N) in the PTR-ToF-MS instruments played a key role in decomposition, not only due to water loss but also carbon-oxygen bond breakage of the detected molecules. Even though PTRMS is considered a soft ionization technique compared to e.g. AMS, these decomposition pathways could still lead to misidentification of the original chemical composition of the SOA species. For the ACM the ionic fragmentation for the gas- and particle-phase species was identical since both measurement were conducted using the same PTR-ToF-MS as a detector. CHARON and TD saturation mass concentration (C*) was determined by using the gas-phase mass concentration measurements derived from a separately deployed PTR-ToF-MS operated at different E/N conditions (see Section 2.2). Ionic dissociation was thus different for the gas- compared to the particle-phase measurements increasing the uncertainty of the volatility estimation for CHARON and TD when compared to ACM.

A method to identify the ionic and thermal dissociation processes and their effect to the different techniques is presented in the following. This method was applied to the calculated average log(C*) of each ion, found both in the gas- and particle-phase, for each experiment for the individual instruments. A characteristic example of the β-pinene ozonolysis experiment (as shown in Figure 1) for the ACM is used here to explain this method. Information of the carbon (x-axis) and oxygen (size of the markers) atom number contained in the chemical formulas were used to differentiate between the different ions

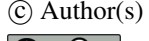



(Figure 1a). Each marker indicates one ion, therefore for the β-pinene experiment 72 ions were detected both in the gas- and
particle-phase by the ACM. Their average saturation mass concentration log(C*) and therefore their volatility ranged from 1
to 4, an indication of semi-volatile and intermediate-volatile species in the SOA mass. From these ions, 55 were identified as
fragmentation products accounting for 70 % of the partitioning ions and only 25 % of these ions were used for further
analysis.
Two major criteria were applied to differentiate between a possible parent ion (green markers) and a fragment. The first
criteria was if the carbon and oxygen atom number were lower than 6 and 1, respectively. This criteria was chosen based on
Donahue et al. (2006) who have shown that organic aerosols are expected in the range from ELVOC to SVOC and IVOC
with saturation concentrations ranging from -5 to 4. This volatility regime consists of species with carbon and oxygen atom
numbers equal or larger than 6 and 1, respectively (Donahue et al., 2011; Donahue et al., 2012). Ions found in the particle-
phase with lower carbon and oxygen numbers were thus considered fragmentation products (grey markers) and were not
used in the analysis. The second criteria focused on the dependence of the volatility to the number of oxygen and carbon
atoms that consitute an organic molecule. As the oxygen and carbon atom number and thus the functionality of the molecule
increased, the saturation mass concentration was expected to decrease (James F. Pankow and Barsanti, 2009). If the volatility
of an identified ion $[M+H]^+$ was identical to (within an uncertainty of log(C*) ± 0.25) or higher than the volatility of ions
with the same chemical formula subtracting a functional group $[M+H-FG]^+$ the latter were considered highly affected by
either ionic or thermal dissociation and were excluded from further analysis. Characteristic examples of this analysis are
shown in Figure 1b and c. The y-axis corresponded to identified ions $[M+H]^+$ while the x-axis to ions with the same
chemical formula subtracting water ($-H_2O$) (Figure 1b) or a carbonyl group ($-CO$) (Figure 1c). When the ions $[M+H]^+$ and
$[M+H-FG]^+$ were found to have identical saturation concentrations, $[M+H-FG]^+$ ions were excluded (blue and orange
markers in Figure 1b and c, respectively). $[M+H-FG]^+$ ions that showed lower volatility when compared to $[M+H]^+$ ions
were considered fragments of unknown decomposition pathways (i.e. unknown parent ion composition) and were excluded
as well (yellow markers). Only when ions $[M+H-FG]^+$ showed higher volatility values than $[M+H]^+$ they were considered to
be possible parent ions not strongly affected by thermal or ionic dissociation (green markers) and were further analyzed. The
same comparison was not only performed for ($-H_2O$) and ($-CO$) functional group loss but was extended to ($-CO_2$), ($-H_2O_2$), ($-
H_2O$) plus ($-CO$), and ($-H_2O$) plus ($-CO_2$). It should be noted that $[M+H]^+$ ions could result from the decomposition of
accretion reaction products or oligomers, consequently leading to an overestimation of their particulate phase concentrations.
This effect is not constrained by this method and is further addressed in Section 3.4. Furthermore, although this method can
efficiently eliminate possible fragments it does not provide proof that these fragments originate from  the suggested
fragmentation pathways.An overview of the fragmentation identification results of this method for each instrument and
experiment are provided in Figure S2. Percentages are derived based on the total number of fragment ions and how they
distribute (%) to the different fragmentation pathways. For all PTR based techniques 40 to 60% of the partitioning ions were
detected below the carbon and oxygen atom number threshold of C6 and O1 respectively. From the remaining species, ions
affected by water ($-H_2O$) loss were around 5-10%, while carboxyl group ($-CO_2$) fragmentation was identified for less than



10% of the partitioning ions. Loss of (-CO), (-$H_2O_2$), (-$H_2O$) plus (-CO) and (-$H_2O$) plus (-$CO_2$) functional groups affected
less than 5% of the ions for all experiments and instruments studied. Ions of unknown decomposition pathways represented
≤ 10% with TD showing the highest values. ACM showed increased contributions of lower molecular weight ions, compared
to TD and CHARON, for limonene and mixture experiments (max 65%). In total, the fraction of ions identified as parent
compounds partitioning in the gas- and particle-phase that were chosen for further analysis in the next sections ranged
between 20-40% of the overall ions found in both phases, for each experiment and instrument studied.
The high contribution of lower MW ions found both in the gas- and particle-phase for all PTR based techniques further
promoted that ionic and thermal dissociation played a key role in carbon-oxygen bond breakage. The higher E/N values of
ACM and TD compared to CHARON resulted in higher fragmentation, thus higher contribution of the lower MW
partitioning ions (Gkatzelis et al., 2017). Although ACM was operated al lower E/N conditions compared to TD, the
contribution of lower MW ions was higher. The reason for this discrepancy was due to the higher limit of detection of the
ACM (section 2.2) compared to TD and CHARON. Ions of low concentrations in the higher MW range that could be
detected from CHARON and TD were below the detection limits of the ACM and were therefore not identified. For the
remaining higher MW species, the water (-$H_2O$) loss was the dominant fragmentation pathway for all techniques. Although
the PTR-based techniques were operated at different temperature, desorption residence times and pressure conditions they
showed similar percent of ions affected by water loss. This is an indication that for all techniques dehydration occurred
mostly due to ionic fragmentation in the ionization region of the PTRMS and not due to thermally initiated reactions for the
partitioning ions studied. TD showed a higher contribution of fragments of unknown decomposition pathways when
compared to ACM and CHARON due to the highest difference of E/N operating conditions in the particle-phase (160 Td)
compared to the gas-phase (120 Td), with the latter measured by a separately deployed PTR-ToF-MS. The higher ionic
dissociation in the particle-phase increased the concentration of lower MW ions and decreased that of higher MW ions. This
had a direct effect on the calculation of the volatility based on equation 2. When this effect was strong enough fragment ions
[M+H-FG]$^+$ showed higher concentrations in the particle phase thus lower volatility when compared to possible parent ions
[M+H]$^+$. These ions were, based on this method, excluded as fragments of unknown fragmentation pathways and showed an
expected higher contribution for systems like the TD. Fragment loss of (-$CO_2$), (-CO), (-$H_2O_2$), (-$H_2O$) plus (-CO) and (-
$H_2O$) plus (-$CO_2$) accounted for 10% or less suggesting that these pathways were not dominating the partitioning ions
studied. Interference of accretion reaction products or oligomers which could be detected at a lower m/z due to
decompostition processes are not accounted for in the precious described method. Possible effect of such an interference is
further discussed in Section 3.4.

**3.2 Volatility distribution coverage**
The mass concentrations of only the species identified as parent ions for ACM, CHARON and TD were distributed to
different volatility bins ranging from log(C*) of -1 to 5 with a 0.5 volatility resolution. The normalized volatility distribution



(NVD) for each experiment accounting for all PTR-based techniques is shown in Figure 2 (1a, 2a, 3a, 4a). Normalization
was performed by dividing each volatility bin by the sum of the PTR-based technique mass concentration measured at each
experiment. The detected biogenic SOA partitioning species showed log(C*) values from 0 to 4, an indication that mainly
SVOCs and IVOCs were predominantly measured simultaneously in the gas- and the particle-phase. The limonene $NO_3$
oxidation experiment had the lowest NVD starting from a log(C*) of 0.5, with a narrow spread up to 2. For the β-pinene and
β-pinene/limonene mixture experiments the NVD moved towards more volatile species ranging from 0.5 to 4. When
comparing the single compound experiment of β-pinene to the mixture, the latter showed a NVD shifted to lower saturation
concentrations. Partitioning species detected from all the PTR-based techniques were further compared as seen in Figure 2
(1b, 2b, 3b, 4b). ACM and CHARON showed same volatility values for all experiments with only the trees experiment
resulting in higher deviations from the one to one line. TD presented higher log(C*) when compared to CHARON and
ACM, suggesting the examined species were underestimated in the particle-phase. A total of 5, 2, 6 and 4 ions were
observed to partition with all three techniques for the β-pinene, limonene, β-pinene/limonene mixture and tree emissions
experiment, respectively after applying the parent ion identification method of section 3.1.
Calculation of the log(C*) in this study relied on the ratio between the gas- and particle-phase signal of an ion (equation 2).
Detection limits of both of these limited the measurable range of this ratio. This explains the narrow volatility distributions
available with all PTR-based techniques, as has been previously reported by Stark et al. (2017). Combining the capabilities
of these instruments and the above approach to calculate the volatility provided insights in a defined range of SVOCs and
IVOCs. Within this volatility range the differences observed when using different precursors agrees with bulk volatility
measurement findings that limonene SOA is less volatile than β-pinene SOA (Lee et al., 2011). When focusing on the
species measured differences of ACM and CHARON to TD could be explained by the higher E/N conditions of TD that
were previously discussed (section 3.1). Since TD was more prone to particle-phase fragmentation compared to the gas-
phase these higher MW compounds showed lower concentrations thus indicated higher volatility. This effect was negligible
for ACM that was using the same PTRMS for gas- and particle-phase measurements and lower for CHARON operated at
lower E/N conditions. The agreement of ACM and CHARON for all experiments except the trees experiment further
promoted that both techniques measured the same species in good agreement and within the uncertainties of these
calculations. As the complexity of the system increased, this agreement deviated from the one to one line. Gkatzelis et al.
(2017) reported that for the single precursor and mixture experiments ions were detected with C6 to C12 carbon atoms from
all techniques. On the contrary, during the tree emissions experiment CHARON was the only instrument to detect ions in the
C13 to C20 range. These ions were not detected from ACM or TD that were operated at higher E/N conditions and were
more prone to ionic and thermal dissociation. Fragmentation of these higher carbon atom ions could affect the volatility
calculation of lower MW species still detected by ACM and TD and thus explain the deviations seen for the tree emissions
experiment.
The total number of species seen from all techniques was low due to the parent ion identification method applied in the
previous section. An overview of the overlapping compounds is provided in Figure S3. When all detected ions were taken



into account more than 50 ions were seen from all techniques at each experiment. After narrowing our focus on the
partitioning ions and excluding the lower MW fragments the overlapping compounds dropped to ~ 15 ions. Each technique
was affected differently by ionic and thermal dissociation. By applying the above method to each technique, different ions
were excluded for each instrument thus leading to only a few species seen from all three techniques and accounted as parent
ions.

**3.3 Experimentally derived saturation concentration implemented to the 2D-VBS**
Species identified as parent ions from each technique were combined and further analysed with a focus on their average
saturation concentration as seen in Figure 3. For parent ions measured from more than one instrument, the average of all
techniques was used to determine the overall experimental C* of the ion, with the error bars indicating the error of this
average. The 2D-VBS (Donahue et al., 2011; Murphy et al., 2012) framework was used to implement the results for each
experiment with background colors corresponding to the different volatility classes, ranging from IVOCs (grey) to SVOCs
(green) and LVOCs (red). It should be noted that the oxidation state (OS) was not derived by bulk measurements using e.g.
the AMS, but by using the OS of the individual species based on their carbon, hydrogen and oxygen atom number (Kroll,
2011). In total 48, 31, 46 and 79 ions were identified as parent ions for the β-pinene, limonene, β-pinene/limonene mixture
and tree emissions oxidation experiment, respectively. The saturation concentration showed a decrease for species with
higher OS and oxygen atom number. For the limonene experiment lower saturation concentration values for compounds
defined by the same oxidation state was found when compared to the β-pinene, mixture or tree emissions experiment.
Overall, parent ions corresponded to 20-30 % of the overall organic mass measured from an AMS for all systems studied.
The observed volatility decrease with increasing OS and oxygen atom number is in good agreement with previous findings
(Jimenez et al., 2009; Kroll, 2011). Lower volatility values for limonene species with the same OS when compared to the β-
pinene, mixture or the tree emissions experiment suggested that species originating from different precursors and oxidation
pathways with differences in their functionality and molecular structure affected their gas-to-particle partitioning. It should
be noted that the lower volatility of limonene could be partly explained by the absence of TD measurement in this
experiment and thus the absence of TD C* values when averaging the experimental results from all PTR-based techniques.
Since TD was affected the strongest by ionic dissociation (highest E/N), the C* values were biased to higher volatilities
when compared to ACM and CHARON with particle-phase measurements ($P_i$ in equation 2) fragmenting more compared to
the gas-phase ($G_i$ from dedicated gas-phase PTR operated at lower E/N). Results when averaging all experiments and
excluding the TD data are shown in Figure S4. Although the limonene experiment would still show lower volatilities
compared to the β-pinene and mixture experiments this trend would be less strong suggesting that the absence of TD during
the limonene experiment did lower the overall average volatility calculation presented in Figure 3. The increased number of
species detected during the tree emissions experiment occurred due to the higher complexity of this system with more than
one precursor oxidized to form SOA. In total, the PTR-based techniques showed that 20-30 % of the overall BSOA mass





consisted of compounds with volatilities within the SVOC to IVOC range further showing the importance of understanding
the gas-to-particle partitioning and thermodynamic properties of compounds formed in such systems.
At this point, it should be noted that losses of gas-phase compounds through the lines, from the SAPHIR to the PTR-MS,
could also affect the log($C^*$) calculation, by changing the ratio of the gas- to the particle-phase. Gas-phase measurements
were performed using a standalone PTR-MS for TD and CHARON while for the ACM both gas- and particle-phase
measurements were obtained using the same PTR-MS of ACM. The two PTR-MS differed in inlet length, temperature, and
material with ACM-PTR-MS introducing higher residence times, thus longer exposure of the gas-phase compounds to the
line walls (see Section 2.2). If significant losses of gas-phase compounds in the ACM-PTR-MS compared to the standalone
PTR-MS line would occur, the gas-phase concentration would be underestimated and therefore also the log($C^*$) derived by
the ACM measurements. To test if the dissimilarities between the different PTR-MS inlet lines are biasing the results of the
ACM, re-calculation of the log($C^*$) was performed by using equation 2 and applying the ACM particle-phase concentration
($P_i$), but changing the gas-phase concentration ($G_i$) to measurements from the standalone PTR-MS. This calculation was
performed for all ions identified as parent ions for the ACM when using the parent ion identification method. An overview
of the correlation of the log($C^*$) using the two different gas-phase datasets is shown in Figure S5. For all experiments and for
most of the compounds, agreement within the uncertainty of the measurements was found. For the tree emissions oxidation
experiment the fraction of compounds deviating from the one to one line was higher. The spread in the data around the one
to one line can be explained by the fact that though both PTR-MS were the same model differences in the design e.g. the
TOF interface existed. These differences introduced additional fragmentation and affected the resolution of the PTR-MS
(Gkatzelis et al., 2017) and could threfore explain the deviations observed. However the differences are within the
experimental uncertainties and therefore no significant bias due to potential inlet line interfence could be determined.

**3.4 Experimentally derived saturation concentration compared to explicit methods**
In order to derive further information from the experimentally determined parent ions, comparison to previous publications
was performed for the major oxidation products from (a) the β-pinene ozonolysis (Chen and Griffin, 2005; Hohaus et al.,
2015; Jenkin, 2004; Kahnt, 2012; Steitz, 2010; Yu et al., 1999), (b) limonene ozonolysis and $NO_3$ oxidation (Chen and
Griffin, 2005; Jaoui et al., 2006; Kundu et al., 2012; Leungsakul et al., 2005a; Leungsakul et al., 2005b) and (c) tree
emissions ozonolysis with α-pinene and $\Delta^3$-carene being the major reactants (Chen and Griffin, 2005; Praplan et al., 2014;
Yu et al., 1999). Species detected as parent ions that overlapped with compounds from previous publications were further
examined based on their structural information. An overview of the overlapping compounds and their suggested structures
are given in Table S1.
A detailed analysis of the β-pinene ozonolysis experiment was performed as seen in Figure 4. Experimental calculation of
the saturation concentration was performed based on the average $C^*$ values throughout the experiment when taking into
account all PTR-based techniques with the error bars indicating the ± 1σ of this averaging. The theoretical calculations by



UManSysProp facility showed that the combinations of the boiling temperature ($T_B$) prediction using NN with the $p_{i,L}$
empirical expressions using MY yielded the maximum C* values while $T_B$ by JB with $p_{i,L}$ by NN yielded the minimum C*
values  (Figure S6). The method originally proposed by Joback and Reid, 1987 to predict boiling points based on the
molecular structure of the investigated compounds explicitly treats ring increments, which are relevant to monoterpene
calculations and thus for this study. Nannoolal et al., 2004  extended the investigated range of functional groups,
simultaneously introducing information on a greater neighborhood of the central atom of the investigated functional group.
The $T_B$ function fitted to the chosen experimental dataset -enlarged as well - yielded lower boiling points for the compounds
investigated here, associated with higher vapor pressure. The method developed by Myrdal and Yalkowsky, 1997 includes
heat capacity changes for phase transitions into their empirical representation, yielding a lowering in the vapor pressure
estimates, compared with the approaches used hitherto (Camredon et al., 2010). The dependency of $\Delta C_p$ upon molecular
flexibility, i.e. the number of torsional bonds (nonterminal $sp^3$ and $sp^2$, rings), makes this inclusion very interesting for
monoterpene calculations. Nannoolal et al., 2008 accounted for the heat capacity changes upon vaporization, too. The new
feature here is that non-additive interaction contribution of multi-functional groups (e.g OH-ketone) are adopted, resulting in
lower vapor pressure values compared with the previous methods. Higher electron delocalization induce stronger dispersive
forces, thus decreasing the $p_{i,L}$. This might explain the larger discrepancy between the vapor pressure values calculated by
NN/MY and JB/NN with the increasing of alcohol/carbonyl/carboxyl functional group number. The methods with the
smallest and largest C* values for the given compounds were chosen to represent the upper and lower limits of the possible
theoretical values, when comparing to the observed ones. These limits are expressed in Figure 4 by the error bars on the x-
axis with the marker points corresponding to their average. In total, 10 compounds were identified from previous
publications to overlap with experimentally detected parent ions for the β-pinene ozonolysis experiment. For most of these
compounds theoretical and experimental values agreed within the uncertainties. No significant discrepancies were found for
compounds in the SVOC volatility range. However, compounds in the IVOC range were underestimated from the
experimental approaches when compared to theory. A characteristic IVOC 1[st] generation product from the β-pinene
ozonolysis is nopinone that has been previously experimentally studied with a focus on the gas-to-particle partitioning
(Hohaus et al., 2015; Kahnt, 2012; Steitz, 2010). Comparison of this work to previous studies was performed as it can be
seen in Figure 4(a). The results showed agreement of the C* within $\pm 10^{0.5}$ between the experimental approaches while the
theory showed differences of $10^3$ in the C* estimation. This comparison was extended to oxonopinone, being the second
underestimated IVOC 1[st] generation product, where again this study (log(C*) = 3.16 ± 0.13) was in good agreement to
Hohaus et al. (2015) (log(C*) = 3.16 ± 0.12 ) using GC-MS but the same sampling technique.
To better understand the differences of the experimental to the theoretical approaches, focus was given on the potential
sources of uncertainties within both calculations. For the theoretical approach, the more complex the molecules with
increasing functional groups were, the higher the uncertainty of the saturation vapor pressure and thus the volatility was.
This is depicted by the higher error bars when moving towards SVOCs. First generation products like nopinone are not



characterized by high complexity, thus theory provided more reliable thermodynamic values also reflected by the good
agreement between all theoretical approaches (Figure S6). The experimental calculation of the volatility performed by the
PTR-based techniques could be affected by the (i) existence of isomers within a studied m/z with different structural
information and thus thermodynamic properties, (ii) thermal and ionic fragmentation of higher molecular weight compounds,
produced by accretion and oligomerization reactions, down to the m/z detected by the instruments, (iii) phase-state of the
bulk OA influencing the partitioning equilibrium time-scales ($\tau_{eq}$) of the individual compounds.
Mass spectrometric measurement approaches provide by definition molecular formulas; however, a given formula does not
correspond to an individual compound. Isaacman-VanWertz et al. (2017) showed that during the α-pinene OH oxidation
molecules with larger carbon atom numbers (C8 to C10) corresponded to an increased number of unique isomers for each
molecular formula. Differences in the functionality of these isomers may be critical for studies of their thermodynamic
properties. To reduce biases in this work, the different isomers seen from previous publications were included in the
theoretical calculations. For the β-pinene experiment the isomers showed C* values within the estimated uncertainty thus
biasing to low extent this comparison. For formulas that corresponded to an individual compound like e.g. nopinone and
oxonopinone further comparison to previous publications was performed. The experimentally calculated C* was in good
agreement with previous studies using a GC-MS to detect particle-phase nopinone (Hohaus et al., 2015; Kahnt, 2012). Since
GC-MS techniques are capable of providing the exact molecular structure of nopinone this further supported the
identification of $(C_9H_{14}O_1)H^+$ and $(C_9H_{12}O_2)H^+$ as protonated nopinone and oxonopinone, respectively, in this study.
The treatment of the PTR dataset to exclude ions affected by thermal and ionic dissociation was described in detail in
section 3.1. However, higher MW species (e.g. accretion reaction products or oligomers), of low volatility, which are not in
the detection range of the PTR-ToF-MS, could decompose to lower MW species during thermal breakdown (Barsanti et al.,
2017) (Tillmann et al., 2010). These species could be identified as a parent ion when using the parent ion identification
method (section 3.1) consequently inducing an overestimation of their particulate phase concentrations. This effect is not
constrained in the used method and could potentially and selectively decrease the volatility of certain species. To explain the
differences in the C* experimental vs theoretical estimations for nopinone, the ratio $\frac{G_i}{P_i}$ from equation 2 should change by a
factor of ~ 300. This would suggest a particulate-phase mass concentration 300 times lower than the observed one, in order
to reach an agreement with the theoretical calculations. This fragmentation pathway should not only strongly affect the PTR-
based techniques but also the previously mentioned GC-MS systems. The decomposition pathway would be narrowed to
thermal dissociation during desorption, which is the only common pathway from all techniques. Finally, this thermal
dissociation pathway needs to result in products with the exact chemical structure of nopinone.
When describing SOA formation, it is generally assumed that oxidation products rapidly adopt gas-to-particle equilibrium
with the assumption of a homogeneously mixed condensed phase (Odum et al., 1996; James F. Pankow, 1994). The non-
ideal behavior of a complex organic mixture could introduce mixing effects, changing the activity coefficients of the
individual organic molecules and thus their gas-to-particle equilibrium. Isotopic labeling experiments have confirmed that



SOA derived from different precursors will interact in a relatively ideal fashion, thus introducing minor deviations of the
activity coefficient from unity (Dommen et al., 2009; Hildebrandt et al., 2011). Furthermore, Hohaus et al. (2015) showed
that for the β-pinene ozonolysis oxidation products the theoretically estimated activity coefficient values calculated by the
thermodynamic group-contribution model AIOFAC (Zuend et al., 2011) were far from explaining the differences between
theory and observations. These findings further suggest that in this work, gas-to-particle partitioning was not strongly
affected by activity coefficient deviations and thus could not explain the obtained differences. On the contrary, the phase-
state of the bulk OA strongly affects the partitioning equilibrium time-scales ($\tau_{eq}$) ranging from seconds in case of liquid
particles to hours or days for semi-solid or glassy particles (M. Shiraiwa et al., 2011; Manabu Shiraiwa and Seinfeld, 2012).
Biogenic SOA particles have been found to adopt an amorphous solid-, most probably glassy-state (Virtanen et al., 2010).
This amorphous solid-state may influence the partitioning of semi-volatile compounds, hindering the lower volatile species
to leave the particles. Biogenic OA produced in this study would be thus directly affected by high partitioning equilibrium
time-scales leading to increased particulate-phase concentrations of more volatile compounds "trapped" within this glassy-
state of the OA. This would imply a direct decay of their volatility thus explaining the observed lower C* values of the 1$^{st}$
generation products.
A comparison of the observed and calculated C* was performed for all experiments during this campaign as shown in Figure
5. There were 11, 12 and 9 compounds observed in the limonene, terpene and trees oxidation experiments, respectively,
which were described in previous publications. These compounds can be attributed to only 5, 8 and 4 different molecular
formulars (m/z) suggesting an increased number of isomers found within these overlaps. The analysis yielded similar
findings to those from the β-pinene experiment. The comparison between observations and theory showed relatively good
agreement within the SVOC range for most of the compounds, while the C* for compounds expected to be in the IVOC
range was experimentally underestimated, i.e. the measured particle-phase concentrations were higher than those explained
by the equilibrium partitioning theory. When moving from single to multiple precursor experiments e.g. from the ozonolysis
of β-pinene to the ozonolysis and NO$_3$ oxidation of limonene, the number of isomers increased rapidly, due to the higher
complexity of the investigated systems. Certain isomers showed variations up to two orders of magnitude in their estimated
volatility values. On the other hand, due to increased complexity of the systems, the limitations of the mass spectrometric
techniques to define the molecular structure of the compounds might introduce large biases. However, despite these
uncertainties, the theoretical volatility values were still found to be in fair agreement with the observations for all systems
studied, suggesting that these deviations would still be within the already existing high uncertainties associated to the
theoretical calculations.
There are two major effects that could be emphasized by presenting two case scenarios. In the first scenario the equilibrium
partitioning theory correctly represents the studied systems. The experimental underestimation of the IVOCs (and certain
SVOCs) volatility can thus only be explained by experimental uncertainties due to (i) fragmentation of higher MW
compounds and oligomers to the detection range of the PTR-based techniques, and/or (ii) the existence of isomers with high
volatility differences. However as mentioned before, studies which performed molecular identification of compounds (e.g.





nopinone) show significantly different experimentally derived partitioning coefficient values when compared to theoretical
calculations (Hohaus et al., 2015; Kahnt et al., 2012), therefore isomers could not explain this discrepancy for all cases. In
the second scenario the assumption of equilibrium partitioning would be questioned due to the findings that BSOA form a
glassy phase-state and thus gas-to-particle equilibrium might not be reached. This would imply that all theoretical
calculations performed in this study and used in models to describe SOA formation would be developed under the wrong
assumption, thus decreasing their reliability. This work provides clear evidence pointing towards these two effects but cannot
provide a quantitative estimate to their individual contribution. Future studies combining the information provided by the
PTR-based techniques with SOA phase-state measurements are essential. In order to bridge the gap between experimental
data and theoretical volatility calculations further development of instrumentation providing structural information at a
molecular level is required. Techniques like the TAG (Isaacman et al., 2014; Williams et al., 2006; Zhang et al., 2014)
coupled in parallel to the PTR-based techniques could provide further insight into different isomeric structures.

**4    Summary**
We have presented the first laboratory inter-comparison of three in-situ, near real-time measurement techniques of gas-to-
particle partitioning with a focus on biogenic SOA formation and oxidation. These thermal desorption techniques are known
to be affected by thermal dissociation during desorption and ionic dissociation during ionization in the drift tube of the
PTRMS (Gkatzelis et al., 2017). These fragmentation pathways could directly affect the gas-to-particle partitioning and thus
the saturation mass concentration (C*) calculation. To reduce fragmentation biases a method to identify and exclude ions
affected by these decomposition pathways was developed and applied. Narrow volatility distributions were observed ranging
from 0 to 4 with species in the semi-volatile (SVOCs) to intermediate volatility (IVOCs) regime. The limonene oxidation
experiment showed a lower volatility distribution when compared to the β-pinene oxidation experiment further supporting
that limonene SOA are less volatile than β-pinene SOA (Lee et al., 2011). When comparing C* values obtained for species
observed from all techniques, instruments showed good agreement within 1 decade, with deviations explained by the
different operating conditions of the PTRMS (Gkatzelis et al., 2017).
Determined species were mapped onto the 2D-VBS framework and results showed a decrease of the C* with increasing
oxidation state and increasing oxygen atom number in accordance to previous findings (Jimenez et al., 2009; Kroll, 2011).
These species accounted for 20-30 % of the total organic mass measured from an AMS. For species that overlapped with
compounds from previous publications a comparison to theoretical calculations was performed based on their molecular
structure. Accounting for the uncertainties of the measurements, results showed good agreement for SVOCs, while IVOCs
introduced higher deviations. Detailed comparison of the partitioning values of nopinone, a 1[st] generation product from the
ozonolysis of β-pinene, was performed to previous publications. Results showed agreement of the C* within $\pm 10^{0.5}$ between
all experimental approaches while theory showed differences of $10^3$ on the C* estimation. These major differences are




discussed in terms of possible uncertainties biasing the experimental values from (1) existence of isomers within a studied m/z, (2) thermal and ionic fragmentation of higher molecular weight compounds, produced by accretion and oligomerization reactions, fragmenting to *m/z*'s detected by the instruments, (3) Non-idealities of the organic mixtures and (4) the phase-state of the bulk OA affecting the partitioning equilibrium time-scales ($\tau_{eq}$) of the individual compounds. Results point towards possible interferences by thermal and ionic fragmentation as well as kinetic influences in the distribution between gas- and particle-phase with diffusivity in the particle-phase and irreversible uptake. These findings further promote future work and parallel measurement of the phase-state of the OA combined with compound specific volatility determination from the PTR-based techniques.

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




**Table 1: Experimental conditions during each ozonolysis experiment. Two VOC injection periods were performed for the tree emissions experiment.**

| Experiment | Ozone (ppbV) | Monoterpenes (ppbV) | Duration (h) | Maximum SOA formed ($\mu g/m^3$) | Chamber temperature (°C) | SOA aging Conditions |
|---|---|---|---|---|---|---|
| **β-Pinene** | 700 | 120 | 34 | 130 | 20 ± 4 | Photochemical oxidation for 10 h |
| **Limonene** | 150 | 25 | 17 | 50 | 17 ± 4 | Continuous $NO_3$ oxidation for 8 h |
| **β-Pinene/Limonene mixture** | 300 | 60/12 | 26 | 60 | 19 ± 5 | Photochemical oxidation for 4 h |
| **Tree emissions** $1^{st}$ inj. / $2^{nd}$ inj. | 300 | 65/10 | 30 | 80 | 30 ± 5 | Photochemical oxidation for 6 h |












**Figure 1: Characteristic example of fragment identification method from the β-pinene ozonolysis experiment for the ACM where (a) is the experimental saturation concentration (y-axis) for all identified compounds with different carbon (x-axis) and oxygen atom number (size of markers). Different colors indicate whether the compound represents a possible parent ion (green), a fragment with carbon and oxygen atom number lower than 6 and 1 respectively (grey), or a fragment originating from the loss of water (blue) or CO (orange). This attribution results from Figure (b) and (c) which show the correlation of the saturation concentration of identified [M+H]$^+$ ions to compounds with the same chemical formula subtracting water [M+H-H$_2$O]$^+$ or CO [M+H-CO]$^+$. If the correlation is close to the 1:1 line then the [M+H-H$_2$O]$^+$ or [M+H-CO]$^+$compound is identified as a fragment and is given the respective color (blue or orange). If the [M+H-H$_2$O]$^+$ or [M+H-CO]$^+$compound shows a higher volatility it is considered as a possible parent ion (green). The orange background indicates the ± 0.25 change of log(C*). Error bars correspond to the error of the average (± 1σ).**





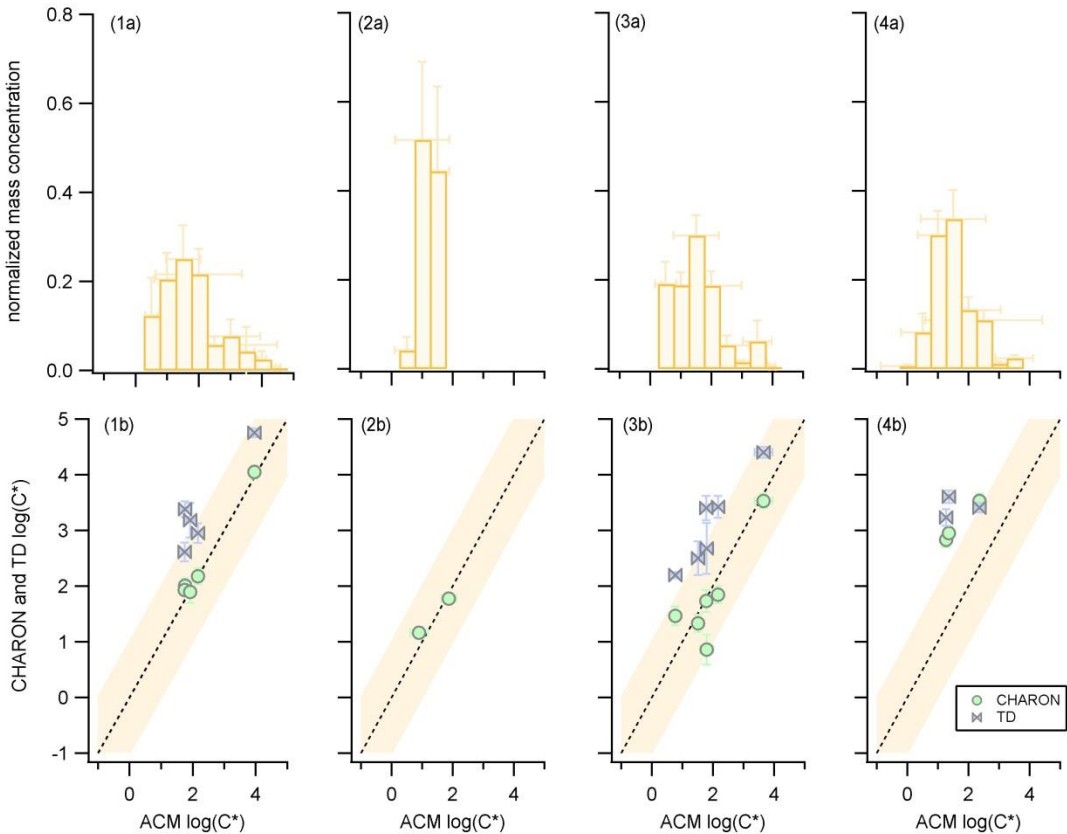

**Figure 2: Oxidation experiments using as precursor (1a,b) β-pinene, (2a,b) limonene, (3a,b) a mixture of β-pinene and limonene and (4a,b) real tree emissions from *Pinus sylvestris L.* (Scots pine). Upper figures (1a, 2a, 3a, 4a) correspond to the normalized average mass concentration from ACM, CHARON and TD, distributed to the different volatility bins with a volatility resolution of 0.5 μg m⁻³. Bottom figures (1b, 2b, 3b, 4b) correspond to the average volatility of overlapping compounds seen from CHARON and ACM (circles) or TD and ACM (double triangles). The dash line represents the 1:1 line. The orange background color indicates the ± 1 μg m⁻³ deviation from the 1:1. Error bars correspond to the ± 1σ of the average throughout each experiment.**





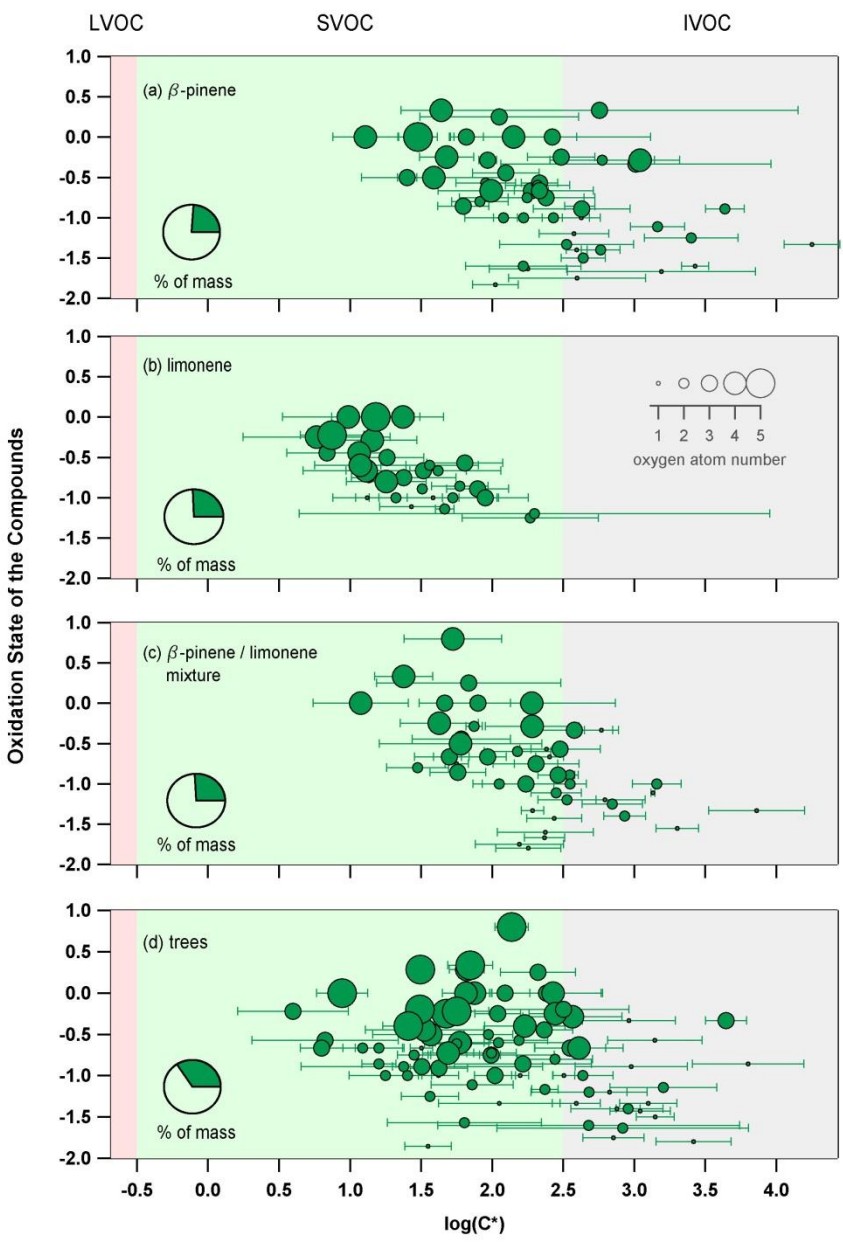

1007

**Figure 3: The average experimental saturation concentration for detected ions (from ACM, CHARON or TD) that act as parent ions identified using the described selection criteria during the (a) β-pinene, (b) limonene, (c) mixture of β-pinene and limonene and (d) the real tree emissions experiments. Error bars indicate the ± 1σ of the experimental average. Size of the markers are an indicator of the oxygen atom number for each species. Pie charts show the percent of mass (green) measured when adding all presented ions compared to the total organic mass obtained from the AMS.**









**Figure 4:** The experimental average saturation concentration obtained from all PTR-based techniques (y-axis) compared to the theoretical calculation of the saturation concentration (x-axis). Theoretical calculations were performed by assuming a chemical structure for the experimentally observed ions. The chemical structure was attributed based on known oxidation products of the β-pinene ozonolysis experiment and are shown on the right side of the figure. Error bars on the y-axis indicate the ± 1σ error of the average based on the experimental results from ACM, TD and CHARON. The error bars for the x-axis act as indicators of the minimum and maximum range of 9 different theoretical approaches with the position of the marker indicating the average of these minimum and maximum values. More details on the theoretical calculations are provided in section 2.4. Sub-figure (a) provides experimentally determined values of the saturation concentration for nopinone based on Hohaus et al. (2015) and Kahnt (2012) together with the results of the experimental and theoretical approaches from this study.








**Figure 5: The experimental average saturation concentration obtained from all PTR-based techniques (y-axis) compared to the theoretical calculation of the saturation concentration (x-axis) for the (i) β-pinene, (ii) limonene, (iii) mixture of β-pinene and limonene and (iv) the real tree emissions experiments. Error bars on the y-axis indicate the ± 1σ error of the average based on the experimental results from ACM, TD and CHARON. The error bars for the x-axis act as indicators of the minimum and maximum range of 9 different theoretical approaches with the position of the marker indicating the average of these minimum and maximum values.**