# Peer review of "Gas-to-particle partitioning of major biogenic oxidation products: A study on freshly formed and aged biogenic SOA"

_Atmospheric Chemistry and Physics, 2018_

## Referee Comment (RC1) · Anonymous Referee #1 · 2 Mar 2018

The authors present new direct measurements of the volatility of biogenic oxidation products, an active area of research in the field, using recently developed inlets for PTR-MS. The major foci of this work are the description of an approach for identifying fragments in PTR-MS data based on measured volatility, demonstration and intercomparison of 3 sampling systems for PTR-MS, and comparison of measured volatility to theoretical. The conclusions are well supported by the results and the recent literature, and the authors are clearly knowledgeable. The work is robust and valuable, and is in general presented clearly. The authors do a nice job of discussing some of the details and considering all of the possible sources of uncertainty. Though I think there are parts of the discussion and analysis that need to be refined and addressed, as indi-

cated below, none are too scientifically substantial - only minor revisions are necessary for publication.

That said, my major concern is that I'm not completely convinced this publication is the proper journal, and wonder whether a more appropriate venue would be Atmospheric Measurement Techniques. Though some of the final conclusions and summary do get at scientific implications of the work, a substantial focus of the manuscript as currently written is devoted to developing an approach for identifying fragments in an instrument (Fig. 1), comparing between sampling approaches (Fig. 2), and demonstrating the capabilities of the approach (Fig. 3). Consequently, it reads somewhat more like it focuses on the techniques than the implications or results. I think it could go a bit either way, and leave it up to the editor, but would argue that it should perhaps go to AMT instead of ACP.

Major comments: 1. Throughout the discussion and presentation of this work, the authors seem to go back and forth somewhat on the lines between ions and molecules. At some points, the fact that the PTR-MS measures ions without structure is a major focus of the discussion. At others, figures and text seem to imply that this work is measuring one specific molecules. Often the authors later clarify, but it makes the discussion somewhat "blurry." As an example, the inset of Figure 4 strongly implies they are measuring nopinone, the fact that it is actually an ion with suggested by unknown structure is not discussed until 40 lines later and not mentioned in the figure. I recommend the authors state clearly when they are discussing ions they measured, vs. when they are discussing specific molecules, and generally shore up their language around these issues. It is a major point in comparing measured vs. theoretical volatility, which they acknowledge, but only really discuss near the end of the manuscript. I think a little reorganization would probably address the issue.

This also confuses the assumptions around fragmentation, since without structure their assumptions that loss of an e.g. C and O atom will increase volatility are not necessarily true (see below). I think their case could still be made, but it needs to be made a bit

more carefully, and the fact that PTR measures ions, not structures is part of that discussion.

2. Comparisons between the ACM, TD, and CHARON are a major part of this work, but there is no real description of them. I recognize that the authors cite their previous work(s) for descriptions, but given the role of these comparisons in this work, at least some cursory description should be provided. How do they differ (other than temperatures and pressures)?

3. The authors often paint thermal and ionic fragmentation with one brush. In some ways that makes sense, I understand, but I don't think I fully agree they should be lumped. Thermal fragmentation is measurement of a molecules that was actually present in the particle as part of an accretion product. Ionic fragmentation is measurement of a molecule that was never in the ambient sample. In some cases this is a meaningful distinction. One major example is the decision to call VOCs fragments and exclude them from future analysis, yet IVOCs are included, even though they are almost certainly thermal fragments in many cases. Essentially the authors have decided that VOCs (C<6, O<1) are too small to be in the particle so are referred to as fragments, yet nopinone is included in the discussion, and later said to possibly come from thermal fragmentation. So then, what is the distinction between nopinone, which the authors continue to include, and those ions deemed "fragments" and excluded? Shouldn't VOCs thus be included in all the later analyses, and in the mass pie charts? They were, after all, observed in the particle phase. Or should those IVOCs be colored as fragments in Figure 1?

Also, the described approach to identifying fragments makes sense for ionic fragments, but not for thermal fragments, which may or many not have the same volatility as their parent (which may or may not be measured). Similarly, ionic fragmentation will likely have a relatively small impact on volatility, while thermal fragmentation probably has a substantial one. So this approach captures one pathway but not the other, suggesting discussion would be clearer if these two processes were more distinctly discussed.

Overall, I think thermal and ionic fragmentation should not be treated together, for instance in Figure 1 and in discussion there and elsewhere (e.g. Section 3.1, lines 496, 593). They have different causes and different implications.

Minor comments: 4. There are significant grammatical errors and language quirks that belie the author as not a non-native English speaker. I have flagged many, but probably not all, below.

5. The introduction generally covers the topics, but it seems like often references are a bit out of place, missing, or not quite correct. I have tried to note these cases below.

Technical comments: line 47-48: "a detailed understanding...needs to be well defined" is odd English. Re-word.

line 54: missing comma between "pressures" and "thus"

line 76: "when applicable" can be deleted. Also, some of the cited works are indeed measuring ions, but others are measuring identified compounds, which is a potentially significant distinction as different molecules of the same formula may partition differently (as discussed latter). See Thompson et al. Aerosol Sci. Tech., 2016, doi: 10.1080/02786826.2016.1254719. Also, citations should probably include Zhao et al., ES&T, 2013, doi: 10.1021/es304587x

line 79: For 2D-TAG, a more appropriate citation is probably Goldstein et al. J. Chrom. A, 2008, doi:10.1016/j.chroma.2007.09.094. I note that most TAG applications are not 2D, so it should be specific in the name here.

line 85: For SV-TAG, citation should include Zhao et al., Aerosol Sci. Tech., 2013, doi: 10.1080/02786826.2012.747673

line 86: I don't disagree with the 10-40% estimate, but it should be cited.

line 95: Krechmer is one of many approaches to estimating c*, and in fact is one of the more complex ones. I might also recommend citing Daumit et al., Faraday Disc., doi:

[Figure]

10.1039/c3fd00045a and Li et al., ACP, 2016, doi:10.5194/acp-16-3327-2016. Those two references directly parameterize by formula, which seems to be the reference in this sentence.

line 113: "Deviations...to" is a bit odd. Maybe "deviations between the theoretical and experimental"

line 130: should be "allows experiments to be conducted"

line 149: should be "extent"

line 168: Is it not a problem that the ACM is at sub-freezing temperatures? Does this not result in some gas-phase adsorption? I'm not sure because the description is sparse. Though the instruments are described elsewhere, given that this manuscript focuses in part on intercomparison, it would be helpful to include a few lines of description about each technique.

line 178: Why was the PTR operating differently for each collector? Could this have any impact, or is it calibrated for?

line 187: Do I correctly understand that all gas-phase measurements are thus passed through a filter? If so, I think this could substantially bias the measurements toward removal of gas-phase compounds and so higher measured particle partitioning. How was this checked or corrected for?

line 204: "pptV" here and elsewhere doesn't need to be capitalized

line 206: should read "additional"

line 220: if I'm not mistaken, this equation should use the molecular weight of the absorbing material, not the compound being absorbed. It looks like that is what is done, but is not what is stated in the description.

line 253: relationship between temperature of c* is log-linear, so a deviation of 15 degrees should vary c* by a multiplier, not a specific number (e.g. 10 times, not 10

ug/m3)

line 253: I'm not completely convinced those temperature variations are as negiglible as the authors seem to assume. Take for example the mixture experiment, with SOA concentrations of 60 ug/m3. A compound with a c* of 60 ug/m3 at 25 degrees (e.g a triol with 7 or 8 carbons, based on SIMPOL) has a c* of 10 ug/m3 at 15 degrees, the range of temperatures in the experiment. That is the difference of 0.8 log units, and means that in the experiment it is the difference between half in the particle and 85% in the particle. Not a huge difference, perhaps, but enough to potentially be a source of uncertainty given the error bars on Figure 1, and probably worth exploring and discussing.

line 294-295: it is a bit confusing to say their volatility ranged from 1 to 4. Maybe just say "10ˆ1 to 10ˆ4 ug/m3"

lines 298-304: I agree that ions that small are likely fragments, but this does not mean that larger ions are not, so the cutoff to include above and disregard below feels a bit arbitrary. See general comments above.

lines 307-309: This assumption of functional groups decreasing volatility makes sense for pure components, but disregards potential impacts of structure. For example if the ion C8H12O2 represents a dione, the ion C7H12O could be an alcohol so be held in a polar particle by stronger hydrogen bonding. I'm not totally convinced that without knowing structures the authors can definitively claim that 2 ions that differ by the atoms that could be a functional group will necessarily have a given relationship in volatility. Previous work (e.g. the Isaacman-VanWertz et al., 2017 reference cited in the manuscript) has used correlation of the timeseries of ions to assess potential overlap, could something similar not be included in this analysis to confirm that fragments co-vary in time as well as volatility? Also, how did the authors deal with nitrates, given that some of the ions contain nitrogen, is loss of the nitrate group possible and/or considered?

line 333: "promoted" should be "supported"

line 359: "0.5 volatility resolution" sounds odd. Maybe add "bin" or units

line 381: missing "and"

line 406: the authors are referring here to the oxidation state of the carbon within the compounds, so should use OS_C as the abbreviation

line 442: For tree emissions in Fig. S5, it looks to me like there is signficant bias between the two approaches, not just random noise. Why might that be true for only this experiment? Does this imply anything for the other comparisons?

line 444: remove "existed"

line 461: misspelled "yielded"

lines 461-477: The detailed discussion of differences between vapor pressure estimation techniques do not seem necessary here. While it is a nice overview, it feels fairly tangential to the focus of the paper and could be removed or moved to the SI.

lines 478-479: In the initial discussion of Figure 4, and in Figure 4 itself, the authors seem to imply they are measuring e.g. "nopinone" not just an ion of the same formula. They go on to have a good discussion of this fact, but it should be made clear throughout the discussion and in the figure. (In other words, the inset of Figure 4 is not really an apples-to-apples comparison, which should be clear).

line 504: This approach to constraining the uncertainty due to structure is nice, but it's description is a little unclear. "within the estimated unceratinty" of what - the experimental values, or the theoretical nopinone values?

line 529: misspelled "AIOMFAC"

line 534: particle phase is humidity dependent, these experiments are at 55%, which could sort of go either way, liquid or solid, for instance see Bateman et al., Nature Geo,

2015, DOI: 10.1038/NGEO2599

Figure 2: Re-label as as a-h, not 1a-2d.

Figure 3: I found sizing by oxygen number to be quite confusing. I recognize the utility of it in Figure 1, but in this figure that information is already more or less captured by the axes, and it serves to highlight certain ions for no real scientific reason. Sizing by concentration or not at all might be more appropriate here.

Figure 5: On some monitors, the dashed lines to isomers cannot be seen. Perhaps darken or color them.

---

## Referee Comment (RC2) · Anonymous Referee #2 · 24 Mar 2018

Gkatzelis and coworkers report measurements of gas-to-particle partitioning of products from biogenic oxidation using three recently developed aerosol inlets and parallel gas-phase measurement. They developed an approach for identifying fragments in data due to thermal dissociation and ionic dissociation. The authors did a decent job in discussing C* intercomparison of 3 aerosol sampling systems and comparison of measured volatility to theoretical calculations. This PTR-based technique can be implemented to promote research in this area. The paper is generally well written. It has a heavy focus on techniques and how they affect the results. I understand that more details are in the cited Gkatzelis (2017) and some are discussed later in the Results section. However, I feel the authors should provide a little more information when rele-

vant and/or refer to the specific section that you discussed in more detail (see specific comments).

Specific comments:

Line 82: Should be "known".

Line 86: Where does the 10-40% come from? Please cite relevant references.

Line 173: Why ACM and TD have different final temperature? How does this affect the compounds they measured?

Line 176-183: The PTRs were operated under different conditions. When you calculate C* using G/P ratio measured by different PTRs (e.g., TD for particle, a standalone PTR for gas), how did you treat the different sensitivities? In addition, a very simple schematic in SI, or table, showing all the instruments connected to the chamber might be helpful to readers, since the authors refer to the different operating conditions, inlets etc. frequently in discussion. Some time series examples of each technique in SI, along with Figure S1, will also be useful.

Line 187: It sounds like the PTFE filter is always before the ACM-PTR-MS then how can the ACM collect particles?

Line 252: Please be explicit what typical vaporization enthalpies are.

Line 253: A change of 15 C will result in C* change larger than 10 ug/m3. For example, a compound with a C*=100 ug/m3 at 290 K will have a C*=700 ug/m3 at 303 K. It is worth to consider and discuss this in the following comparison.

Line 286: Can the authors estimate the uncertainty caused by operating gas and particle phase measurements under different ionic dissociation?

Line 317: Did the authors do similar test for organic nitrate products (-HNO3)?

Line 342: Should be "percentage".

Line 354: Should be "previous".

Figure 2: The error bar for the green dots need a darker color since it's hard to see. The ACM C*s don't have any error bar. Why, too small? In addition, since only averages were presented, it'd be worth to mention in section 2.4 that you calculated C* from equation 2 using how many samples for each technique for each experiment. As temperature varied though an experiment, how was C* affected?

Figure S6: Temp = 280K, why inconsistent with 298K used for experimental measurement mentioned at line 256? The name for each theoretical calculation is not consistent with that in the text, e.g., nano vs. NN.

---

## Author Comment (AC1) · 27 Jun 2018

**Answers to anonymous referee #1**

The authors present new direct measurements of the volatility of biogenic oxidation products, an active area of research in the field, using recently developed inlets for PTR-MS. The major focus of this work are the description of an approach for identifying fragments in PTR-MS data based on measured volatility, demonstration and inter-comparison of 3 sampling systems for PTR-MS, and comparison of measured volatility to theoretical. The conclusions are well supported by the results and the recent literature, and the authors are clearly knowledgeable. The work is robust and valuable, and is in general presented clearly. The authors do a nice job of discussing some of the details and considering all of the possible sources of uncertainty. Though I think there are parts of the discussion and analysis that need to be refined and addressed, as indicated below, none are too scientifically substantial -only minor revisions are necessary for publication.

**Answer:** We do appreciate the positive assessment. We have addressed the various comments and suggestion of the referee as described below. Changes in the manuscript are highlighted in *italic*.

That said, my major concern is that I'm not completely convinced this publication is the proper journal, and wonder whether a more appropriate venue would be Atmospheric Measurement Techniques. Though some of the final conclusions and summary do get at scientific implications of the work, a substantial focus of the manuscript as currently written is devoted to developing an approach for identifying fragments in an instrument (Fig. 1), comparing between sampling approaches (Fig. 2), and demonstrating the capabilities of the approach (Fig. 3). Consequently, it reads somewhat more like it focuses on the techniques than the implications or results. I think it could go a bit either way, and leave it up to the editor, but would argue that it should perhaps go to AMT instead of ACP.

**Answer:** We understand the dilemma of the referee given that our study includes both elements of method development and also its application and scientific outcome in chamber studies. However, Fig.3, Fig. 4 and Fig. 5, do show results of direct volatility measurements of major biogenic oxidation products and their comparison to theoretical approaches. To our knowledge these scientific findings represent new valuable insights that extend beyond just the comparison of different atmospheric measurement techniques. We do agree that an essential step in order to achieve a reliable conclusion for this study was the development of a method to exclude possible fragments. Nonetheless, this method has only been used as a tool in order to narrow down the uncertainties and provide more reliable conclusions for the above figures.

**Major comments**:

Throughout the discussion and presentation of this work, the authors seem to go back and forth somewhat on the lines between ions and molecules. At some points, the fact that the PTR-MS measures ions without structure is a major focus of the discussion. At others, figures and text seem to imply that this work is measuring one specific molecule. Often the authors later clarify, but it makes the discussion somewhat "blurry." As an example, the inset of Figure 4 strongly implies they are measuring nopinone, the fact that it is actually an ion with suggested by unknown structure is not discussed until 40 lines later and not mentioned in the figure. I recommend the authors state clearly when they are discussing ions they measured, vs. when they are discussing specific molecules, and generally shore up their language around these issues. It is a major point in comparing measured vs. theoretical volatility, which they acknowledge, but only really discuss near

the end of the manuscript. I think a little reorganization would probably address the issue. This also confuses the assumptions around fragmentation, since without structure their assumptions that loss of an e.g. C and O atom will increase volatility are not necessarily true (see below). I think their case could still be made, but it needs to be made a bit more carefully, and the fact that PTR measures ions, not structures is part of that discussion.

**Answer**: Changes in order to further clarify when the discussion is focused on ions and when the assumption of a chemical structure is implied were performed throughout the text as following:
- For Figure 4 the phrase "*suggested structure*" was added on top of the compounds.
- Line 427: Replaced "compounds" with "*ions*".
- Line 454: Changed "Species detected as parent ions that overlapped with compounds from previous publications were further examined based on their structural information" to "*By attributing a chemical structure to the ions identified by the PTR-MS, detected parent ions that overlapped with compounds from previous publications were further examined based on their structural information*"
- Line 456: Added "*Uncertainties introduced by assigning a chemical structure to an ion of a given chemical formula are further discussed in this section.*"

Comparisons between the ACM, TD, and CHARON are a major part of this work, but there is no real description of them. I recognize that the authors cite their previous work(s) for descriptions, but given the role of these comparisons in this work, at least some cursory description should be provided. How do they differ (other than temperatures and pressures)?

**Answer:** We added a paragraph giving the important difference in the set-up of each instrument to highlight the similarities and difference of the instruments. The changes in the manuscript are as follows:

"*In the following, the most important characteristics and parameters are described briefly. The CHARON inlet combines a gas phase denuder, an aerodynamic lens with an inertial sampler and a thermodesorption unit which is coupled to a PTR-ToF-MS. The gas phase denuder removes gas phase analytes. Subsequently the aerosols are collimated by the aerodynamic lens and a particle enriched sample flow is achieved by the inertial sampler. Afterwards the particles pass through a thermal desorption unit in which the particles are volatilized before transferred to the gas phase detector. The ACM has two sample air inlets. For the gas phase inlet air passes through a PTFE particle filter and is then directly introduced into the PTR-MS. For particle collection via the second sampling line air is passing through an aerodynamic lens removing gas phase and collimating particles onto a beam. The particles are subsequently passing a vacuum chamber and are collected on a cooled sampling surface. Once collection is finished particles are desorbed and transferred via a carrier gas ($N_2$) to the PTR-ToF-MS detector. Important to note is that during the collection process the PTR-ToF-MS is measuring the gas phase in parallel allowing for quasi simultaneous characterization of gas and particle phase. The TD also employs a gas phase denuder to remove gas phase analytes before the aerosols are impacted using a Collection and Thermo-Desorption (CTD) cell. After collection particles are thermally desorbed and the components transferred to the PTR-ToF-MS. In the following, operational parameters are listed for all PTR-based instruments. The CHARON is a real time measurement (10 s integration time in the detector), while the ACM and TD have sampling times for this study of 120 min and 240 min, respectively.*"

and

*"The operational conditions for each PTR-ToF-MS were different with regard to a different electric field strength (V cm$^{-1}$) to buffer gas density (molecules cm$^{-3}$) ratio (E/N). This can lead to different ionic fragmentation behavior. Therefore, the overlap of parent ions measured between the different instruments can be reduced. A detailed discussion about the E/N effect has been investigated by Gkatzelis et al. (2018a). Operational details for the different PTR-ToF-MS conditions are given also in Table S2."*

The authors often paint thermal and ionic fragmentation with one brush. In some ways that makes sense, I understand, but I don't think I fully agree they should be lumped. Thermal fragmentation is measurement of a molecule that was actually present in the particle as part of an accretion product. Ionic fragmentation is measurement of a molecule that was never in the ambient sample. In some cases this is a meaningful distinction. One major example is the decision to call VOCs fragments and exclude them from future analysis, yet IVOCs are included, even though they are almost certainly thermal fragments in many cases. Essentially the authors have decided that VOCs (C<6, O<1) are too small to be in the particle so are referred to as fragments, yet nopinone is included in the discussion, and later said to possibly come from thermal fragmentation. So then, what is the distinction between nopinone, which the authors continue to include, and those ions deemed "fragments" and excluded? Shouldn't VOCs thus be included in all the later analyses, and in the mass pie charts? They were, after all, observed in the particle phase. Or should those IVOCs be colored as fragments in Figure 1? Also, the described approach to identifying fragments makes sense for ionic fragments, but not for thermal fragments, which may or may not have the same volatility as their parent (which may or may not be measured). Similarly, ionic fragmentation will likely have a relatively small impact on volatility, while thermal fragmentation probably has a substantial one. So this approach captures one pathway but not the other, suggesting discussion would be clearer if these two processes were more distinctly discussed. Overall, I think thermal and ionic fragmentation should not be treated together, for instance in Figure 1 and in discussion there and elsewhere (e.g. Section 3.1, lines 496, 593). They have different causes and different implications.

**Answer:** Thermal fragmentation can indeed be a result of already existing molecules that were present in the particle as part of an accretion product but this is not proven to be the only possible fragmentation pathway. Exposure of molecules during desorption to high temperatures for long residence times may initiate the breakage of carbon-carbon or carbon-oxygen bonds thus resulting to a variety of possible fragmentation pathways including the generation of fragment molecules that were not in the ambient sample before. Furthermore, although ionic fragmentation is dominated by the suggested loss pathways included in the method, examples of stronger loss processes do exist. A characteristic example is the detection of monoterpenes. Although PTR-MS detects monoterpenes at m/z 137.13 as $C_{10}H_{16}H^+$, a high signal is also found in m/z 81.07 as $C_6H_9^+$. This difference will result in a relatively large impact on volatility that is directly linked to ionic and not thermal fragmentation. Therefore, a clear separation of thermal and ionic dissociation in these complex systems can be challenging. Nevertheless, although the current state of knowledge does not support a clear separation of thermal and ionic fragmentation, our method is capable of excluding all detectable fragments, independent of their origin. Currently, a detailed lab characterization of the different techniques, at different operating conditions is performed to further address differences and possible thermal and ionic fragmentation pathways.

We do agree with the reviewer that the limits chosen for this method (C<6 and O<1) might be uncertain but based on various publications, that we make the reader aware of in the manuscript, it is indubitable that compounds below this carbon and oxygen atom number, considered VOCs, will be

in the gas-phase. On the contrary, various studies have been performed (Kahnt, 2012;Hohaus et al., 2015) where focus was given on the partitioning of compounds in the IVOC range. These compounds and for example nopinone have troubled the scientific community the last years regarding to why experimental results repeatedly show more in the particle-phase than predicted from theory. A major scientific question and the focus of many recent studies (Shiraiwa et al., 2011) has been towards understanding the phase-state of the particles which could strongly affect the partitioning of compounds in the IVOC and SVOC range. Results from this work further address these questions and limit the possible explanations to either thermal fragmentation or particle phase-state. Excluding completely IVOCs from this method as fragments would imply that we are certain that the phase-state of the particles does not affect the partitioning of these compounds and that thermal fragmentation is the key process, something which is not yet known and could thus bias the results of this study.

**Minor comments**:

There are significant grammatical errors and language quirks that belie the author as not a non-native English speaker. I have flagged many, but probably not all, below.

**Answer:** We have corrected all errors mentioned and improved the language of the manuscript where necessary.

The introduction generally covers the topics, but it seems like often references are a bit out of place, missing, or not quite correct. I have tried to note these cases below.

**Answer:** The introduction was improved according to the suggestions.

**Technical comments**:

line 47-48: "a detailed understanding...needs to be well defined" is odd English. Re-word.

**Answer:** The sentence was changed in the manuscript to:

*A detailed understanding of SOA formation and composition is critical to develop strategies needs to be well defined for impact mitigation.*

line 54: missing comma between "pressures" and "thus"

**Answer:** Added

line 76: "when applicable" can be deleted. Also, some of the cited works are indeed measuring ions, but others are measuring identified compounds, which is a potentially significant distinction as different molecules of the same formula may partition differently (as discussed latter). See Thompson et al. Aerosol Sci. Tech., 2016, doi: 10.1080/02786826.2016.1254719. Also, citations should probably include Zhao et al., ES&T, 2013, doi: 10.1021/es304587x

**Answer:** We added the following two sentences to the manuscript:

*Simultaneous measurements of the gas- and particle-phase mass of organic molecules has also been recently developed using the TAG system sampling alternately with and without a gas phase denuder*

*in front of the inlet (Zhao et al., 2013) and the modified semi-volatile TAG (SV-TAG) that utilizes two TAG cells in parallel (Isaacman-VanWertz et al., 2016).*

and

*Measurements of instruments providing molecular identification (e.g. SV-TAG) and measurements from instruments providing identification of ions (e.g. different Chemical Ionization Mass Spectrometer (CIMS)) can be combined to increase the understanding of partitioning of some compounds classes. This was shown in an field intercomparison investigating gas-particle partitioning of oxygenated VOCs during the Southern Oxidant and Aerosol Study (SOAS) (Thompson et al., 2016).*

line 79: For 2D-TAG, a more appropriate citation is probably Goldstein et al. J. Chrom. A, 2008, doi:10.1016/j.chroma.2007.09.094. I note that most TAG applications are not 2D, so it should be specific in the name here.

**Answer:** Corrected

line 85: For SV-TAG, citation should include Zhao et al., Aerosol Sci. Tech., 2013, doi: 10.1080/02786826.2012.747673

**Answer:** Added

line 86: I don't disagree with the 10-40% estimate, but it should be cited.

**Answer:** Added

line 95: Krechmer is one of many approaches to estimating c*, and in fact is one of the more complex ones. I might also recommend citing Daumit et al., Faraday Disc., doi: 10.1039/c3fd00045a and Li et al., ACP, 2016, doi:10.5194/acp-16-3327-2016. Those two references directly parameterize by formula, which seems to be the reference in this sentence.

**Answer:** Added

line 113: "Deviations...to" is a bit odd. Maybe "deviations between the theoretical and experimental"

**Answer:** Done

line 130: should be "allows experiments to be conducted"

**Answer:** Done

line 149: should be "extent"

**Answer:** Done

line 168: Is it not a problem that the ACM is at sub-freezing temperatures? Does this not result in some gas-phase adsorption? I'm not sure because the description is sparse. Though the instruments are described elsewhere, given that this manuscript focuses in part on intercomparison, it would be helpful to include a few lines of description about each technique.

**Answer:** The ACM inlet is constructed similar to an Aerodyne Aerosol Mass Spectrometer meaning the sampling air passes through an aerodynamic lens and subsequently through a high vacuum chamber before the particles are impacted on a sub-zero cooled collection surface. So the gas phase

is efficiently removed before particle sampling. We added a more detailed description to the instrument to highlight that fact. Please see answer above.

line 178: Why was the PTR operating differently for each collector? Could this have any impact, or is it calibrated for?

**Answer:** The PTR-ToF-MS detector of each instrument was operated during the measurement campaign using best practice by each of the respective groups to achieve optimal measurement results. In a nutshell the PTR-ToF-MS E/N settings have to be set to optimize the conditions balancing sensitivity versus ionic fragmentation. As discussed in detail in Gkatzelis et al. 2018 differences in the measurement results obtained from the different instrument originates predominantly from the detector settings and not from the differences in the aerosol sampling inlets which was unexpected. Therefore, in future studies it is advised to try to reduce the difference in PTR detector operations between the different instruments. However, the major impact for the instruments which were operating at higher E/N settings is that we fragment more molecules during ionization compared to lower E/N settings. That means that we limited our analysis since we excluded any ion signal which originates from ionic fragmentation. It is likely that with the same E/N settings in each instrument we would be able to identify more parent ions showing minor ionic fragmentation. Therefore, it has no impact on the identified parent ions and the main conclusion only on the amount of identified parent ions and the overlap of identified parent ions between instruments.

line 187: Do I correctly understand that all gas-phase measurements are thus passed through a filter? If so, I think this could substantially bias the measurements toward removal of gas-phase compounds and so higher measured particle partitioning. How was this checked or corrected for?

**Answer:** Only the PTR-MS of the ACM was operated by passing the gas-phase measurements through a filter. To make this clear the sentence was changed to *"… was additionally introduced to the PTR-MS line of the ACM to reassure complete particle-phase removal."*
Detailed discussion concerning the differences related to the gas-phase measurements of the two PTR-MS are in lines 430 to 447.

line 204: "pptV" here and elsewhere doesn't need to be capitalized

**Answer:** Changed in line 204, 207, and Table 1

line 206: should read "additional"

**Answer:** Done

line 220: if I'm not mistaken, this equation should use the molecular weight of the absorbing material, not the compound being absorbed. It looks like that is what is done, but is not what is stated in the description.

**Answer:** We thank the reviewer for noticing this error.

line 253: relationship between temperature of c* is log-linear, so a deviation of 15 degrees should vary c* by a multiplier, not a specific number (e.g. 10 times, not 10 ug/m3)

**Answer:** The sentence was changed to *"…with changes of ± 15 °C resulting in a C\* change by a factor of 10."*

line 253: I'm not completely convinced those temperature variations are as negligible as the authors seem to assume. Take for example the mixture experiment, with SOA concentrations of 60 ug/m3. A

compound with a c* of 60 ug/m3 at 25 degrees (e.g a triol with 7 or 8 carbons, based on SIMPOL) has a c* of 10 ug/m3 at 15 degrees, the range of temperatures in the experiment. That is the difference of 0.8 log units, and means that in the experiment it is the difference between half in the particle and 85% in the particle. Not a huge difference, perhaps, but enough to potentially be a source of uncertainty given the error bars on Figure 1, and probably worth exploring and discussing.

**Answer:** We agree that the assumption can be improved and therefore, uncertainties were further examined with a focus on the β-pinene oxidation products. An overview of the temperature dependence was calculated from the theoretical approaches and added as a figure in the supplementary material (Figure S7). Figure S7 was also added at the end to these answers below. These uncertainties were dependent on the structure of the compounds and ranged from 0.3 to 0.6 in log(C*). These uncertainties are discussed in the text as follows:

*"The uncertainty added from these variations (< 10 °C) was further examined with a focus on the β-pinene oxidation products (Figure S7). Difference in volatility due to variations ranged from 0.3 to 0.6 log(C*) units depending on the chemical structure of the compound. Nevertheless, these variations can be considered small and not strongly affecting the conclusions of this work."*

line 294-295: it is a bit confusing to say their volatility ranged from 1 to 4. Maybe just say "10^1 to 10^4 ug/m3"

**Answer:** Done

lines 298-304: I agree that ions that small are likely fragments, but this does not mean that larger ions are not, so the cutoff to include above and disregard below feels a bit arbitrary. See general comments above.

**Answer:** Please refer to the answer to the third major comment.

lines 307-309: This assumption of functional groups decreasing volatility makes sense for pure components, but disregards potential impacts of structure. For example if the ion $C_8H_{12}O_2$ represents a dione, the ion $C_7H_{12}O$ could be an alcohol so be held in a polar particle by stronger hydrogen bonding. I'm not totally convinced that without knowing structures the authors can denitively claim that 2 ions that differ by the atoms that could be a functional group will necessarily have a given relationship in volatility. Previous work (e.g. the Isaacman-VanWertz et al., 2017 reference cited in the manuscript) has used correlation of the timeseries of ions to assess potential overlap, could something similar not be included in this analysis to confirm that fragments covary in time as well as volatility? Also, how did the authors deal with nitrates, given that some of the ions contain nitrogen, is loss of the nitrate group possible and/or considered?

PTR-ToF-MS provides information regarding the chemical formula of a compound and not the chemical structure as would for example a GC-MS. As correctly mentioned, this can lead to the identification of a parent ion as a fragment although it may not be. Nevertheless, although this method will potentially exclude compounds that are parent ions, it will still discard any possible fragments. Correlation analysis based on the time series of the different compounds is a very good way to further improve this method but unfortunately for this work not feasible due to the low time resolution. These points are further discussed in the manuscript as follows:

*"It should be noted that PTR-MS provides information regarding the chemical formula of an ion and thus disregards potential impact of the chemical structure. Functionality effect (e.g. stronger hydrogen bonding of an alcohol in a polar particle) can lead to a misidentification of potential parent ions as a fragment using the above described method due to the fact that a lower volatility is*

*determined compared to an expected volatility based on the chemical formula. Nevertheless, although this method will potentially exclude parent ions, it will still discard also any possible fragments. Correlation analysis based on the time series of the different compounds could further improve the parent ion identification. However due to the low time resolution in this work a time series analysis is not applicable. Another implication relies on the fact that [M+H]$^+$ ions could result from the decomposition of accretion reaction products or oligomers, consequently leading to an overestimation of their particulate phase concentrations."*

Organic Nitrates strongly fragment in the PTR-MS and almost certainly lose their nitrate group during ionization with H3O+ (Marius Duncianu et al., 2017). Since no compound was identified as an organic nitrate for the limonene experiment, this fragmentation pathway was not chosen in this method since there was no larger molecule to compare to. This is now further mentioned in the manuscript as follows:

*"Checks were also performed for loss of the (-HNO3) functional group for the limonene-NO3 oxidation experiment but due to the high E/N operating conditions of all PTR-ToF-MS systems, no organic nitrates were identified."*

line 333: "promoted" should be "supported"

**Answer:** Done

line 359: "0.5 volatility resolution" sounds odd. Maybe add "bin" or units

**Answer:** Added "bin"

line 381: missing "and"

**Answer:** Added.

line 406: the authors are referring here to the oxidation state of the carbon within the compounds, so should use OS_C as the abbreviation

**Answer:** Done.

line 442: For tree emissions in Fig. S5, it looks to me like there is significant bias between the two approaches, not just random noise. Why might that be true for only this experiment? Does this imply anything for the other comparisons?

**Answer:** The reason of this difference is due to the higher complexity of the tree emissions in comparison to the single precursor oxidation experiments. As has been shown from Gkatzelis et al. (2018b), the tree emissions experiment was the only experiment were ions with up to 20 carbon atoms were identified from CHARON when operated at low E/N conditions. These ions fragmented differently in the PTR-MS when operated at different conditions and thus introduced a larger deviation from the one to one line as shown in Fig. S5. Additional information to make this point clear was added in the manuscript:

*"Moreover, the tree emissions experiment showed the highest complexity in comparison to the single precursor oxidation experiments, with detected ions that had up to 20 carbon atoms in the particles. These higher molecular weight ions fragmented differently when passing through the differing ToF interfaces and thus resulted to the observed higher deviation."*

line 444: remove "existed"

**Answer:** Corrected.

line 461: misspelled "yielded"

**Answer:** Corrected.

lines 461-477: The detailed discussion of differences between vapor pressure estimation techniques do not seem necessary here. While it is a nice overview, it feels fairly tangential to the focus of the paper and could be removed or moved to the SI.

**Answer:** These sentences are moved to the SI and a sentence is added saying: *"More details regarding the theoretical calculations are provided in the Supplement."*

lines 478-479: In the initial discussion of Figure 4, and in Figure 4 itself, the authors seem to imply they are measuring e.g. "nopinone" not just an ion of the same formula. They go on to have a good discussion of this fact, but it should be made clear throughout the discussion and in the figure. (In other words, the inset of Figure 4 is not really an apples-to-apples comparison, which should be clear).

**Answer:** Please see answer to the first major comment above.

line 504: This approach to constraining the uncertainty due to structure is nice, but it's description is a little unclear. "within the estimated uncertainty" of what -the experimental values, or the theoretical nopinone values?

**Answer:** The sentence was changed to:

*"For the β-pinene experiment the isomers showed theoretical C\* values within the estimated uncertainty thus biasing to a minor extent this comparison."*

line 529: misspelled "AIOMFAC"

**Answer:** Corrected.

line 534: particle phase is humidity dependent, these experiments are at 55%, which could sort of go either way, liquid or solid, for instance see Bateman et al., Nature Geo, 2015, DOI: 10.1038/NGEO2599

**Answer:** The discussion of the aerosol phase state was put into context to the experimental conditions in this study. The manuscript was revised as follows:

*The experimental conditions in this study (on average 55 % RH) suggest that a significant portion of the SOA can be in a semi-solid or glassy state (Bateman et al., 2015).*

Figure 2: Re-label as as a-h, not 1a-2d.

**Answer:** Corrected.

Figure 3: I found sizing by oxygen number to be quite confusing. I recognize the utility of it in Figure 1, but in this figure that information is already more or less captured by the axes, and it serves to highlight certain ions for no real scientific reason. Sizing by concentration or not at all might be more appropriate here.

**Answer:** We removed the sizing by oxygen for Figure 3 and also for Figure S4 to be consistent.

Figure 5: On some monitors, the dashed lines to isomers cannot be seen. Perhaps darken or color them.

Answer: Corrected.

**References**

Gkatzelis, G. I., Tillmann, R., Hohaus, T., Müller, M., Eichler, P., Xu, K. M., Schlag, P., Schmitt, S. H., Wegener, R., Kaminski, M., Holzinger, R., Wisthaler, A., and Kiendler-Scharr, A.: Comparison of three aerosol chemical characterization techniques utilizing ptr-tof-ms: A study on freshly formed and aged biogenic soa, Atmos. Meas. Tech., 11, 1481-1500, 10.5194/amt-11-1481-2018, 2018a.

Gkatzelis, G. I., Tillmann, R., Hohaus, T., Müller, M., Eichler, P., Xu, K. M., Schlag, P., Schmitt, S. H., Wegener, R., Kaminski, M., Holzinger, R., Wisthaler, A., and Kiendler-Scharr, A.: Comparison of three aerosol chemical characterization techniques utilizing ptr-tof-ms: A study on freshly formed and aged biogenic soa, Atmos Meas Tech, 11, 1481-1500, 10.5194/amt-11-1481-2018, 2018b.

Hohaus, T., Gensch, I., Kimmel, J. R., Worsnop, D. R., and Kiendler-Scharr, A.: Experimental determination of the partitioning coefficient of β-pinene oxidation products in soas, Phys Chem Chem Phys, 17, 14796-14804, 10.1039/C5CP01608H, 2015.

Kahnt, A.: Semivolatile compounds from atmospheric monoterpene oxidation PhD, Fakultät für Chemie und Mineralogie, Universität Leipzig, Leipzig, Germany, 205 pp., 2012.

Shiraiwa, M., Ammann, M., Koop, T., and Pöschl, U.: Gas uptake and chemical aging of semisolid organic aerosol particles, P Natl Acad Sci USA, 108, 11003-11008, 10.1073/pnas.1103045108, 2011.

[Figure]

**Figure S7: Theoretical calculation of the vapor pressure (y-axis) using the combination of 7 different approaches. To estimate the uncertainty in the experiments due to changes in temperature calculations were also performed at 295 K. Difference in vapor pressure are between 0.3 to 0.6 log(C\*).**

---

## Author Comment (AC2) · 27 Jun 2018

**Answers to anonymous referee #2**

Gkatzelis and coworkers report measurements of gas-to-particle partitioning of products from biogenic oxidation using three recently developed aerosol inlets and parallel gas-phase measurement. They developed an approach for identifying fragments in data due to thermal dissociation and ionic dissociation. The authors did a decent job in discussing C* intercomparison of 3 aerosol sampling systems and comparison of measured volatility to theoretical calculations. This PTR-based technique can be implemented to promote research in this area. The paper is generally well written. It has a heavy focus on techniques and how they affect the results. I understand that more details are in the cited Gkatzelis (2017) and some are discussed later in the Results section. However, I feel the authors should provide a little more information when relevant and/or refer to the specific section that you discussed in more detail (see specific comments).

We thank the Reviewer for all the comments. We increased the clarity by adding information regarding the measurement techniques and how the differences between the instruments might and can influence the results. For details please see our answers below and also answers to Reviewer 1.

**Specific comments:**

Line 82: Should be "known".

**Answer:** Corrected.

Line 86: Where does the 10-40% come from? Please cite relevant references.

**Answer:** Please see answer to a similar comment of Reviewer 1.

Line 173: Why ACM and TD have different final temperature? How does this affect the compounds they measured?

**Answer:** Each instrument was operated during the measurement campaign using best practice by each of the respective groups to achieve optimal measurement results based on previous experience. One expected effect is that the ACM might not desorb and detect compounds which are in the additional temperature range covered by the TD. Therefore, the ACM might measure less compounds than the TD but this would have no influence on the results since we restrict the analysis to measured ions which we identify with our method as likely being parent ions.

Line 176-183: The PTRs were operated under different conditions. When you calculate C* using G/P ratio measured by different PTRs (e.g., TD for particle, a standalone PTR for gas), how did you treat the different sensitivities? In addition, a very simple schematic in SI, or table, showing all the instruments connected to the chamber might be helpful to readers, since the authors refer to the different operating conditions, inlets etc. frequently in discussion. Some time series examples of each technique in SI, along with Figure S1, will also be useful.

We included in the SI Table S2 which shows the details of the conditions for every PTR-ToF-MS used in this study. Difference in sensitivities of the two instruments did not affect this comparison but the main source of uncertainty was introduced due to the different E/N operating conditions. This is now discussed in more detail in the manuscript and has been the main focus of our previous publication (Gkatzelis et al., 2018). The following sentence is added in the manuscript and also the suggested

example time series in the SI in Figure S8. Figure S8 was also added at the end to these answers below.

*"Finally, differences in sensitivity for each PTR-MS introduced minor deviations in this study and are discussed in detail in Gkatzelis et al., (2018). A characteristic timeseries of a major oxidation product from the β-pinene ozonolysis for the three different techniques can be found in Figure S8."*

Line 187: It sounds like the PTFE filter is always before the ACM-PTR-MS then how can the ACM collect particles?

**Answer:** The description of all instruments was significantly extended in the instrumental section of the manuscript. For all details please see answer to Reviewer 1. The fact that the ACM measures both, gas- and particle phase, was especially clarified. The relevant changes in the manuscript are as follows:

*"The ACM has two sample air inlets. For the gas phase inlet air passes through a PTFE particle filter and is then directly introduced into the PTR-MS. For particle collection via the second sampling line air is passing through an aerodynamic lens removing gas phase and collimating particles onto a beam. The particles are subsequently passing a vacuum chamber and are collected on a cooled sampling surface. Once collection is finished particles are desorbed and transferred via a carrier gas (N2) to the PTR-ToF-MS detector. Important to note is that during the collection process the PTR-ToF-MS is measuring the gas phase in parallel allowing for quasi simultaneous characterization of gas and particle phase."*

Line 252: Please be explicit what typical vaporization enthalpies are.

**Answer:** Added.

Line 253: A change of 15 C will result in C* change larger than 10 ug/m3. For example, a compound with a C*=100 ug/m3 at 290 K will have a C*=700 ug/m3 at 303 K. It is worth to consider and discuss this in the following comparison.

**Answer:** Please see the answer to Reviewer 1 for the same question.

Line 286: Can the authors estimate the uncertainty caused by operating gas and particle phase measurements under different ionic dissociation?

**Answer:** Ionic dissociation in the PTR-MS detector is compound dependent. Therefore, a general uncertainty cannot be estimated/applied to the different E/N conditions of the PTR detector. Due to the thermal desorption of the particle phase thermal fragmentation occurs additional to the ionic fragmentation for the particle composition measurements. Since it is not possible to distinguish thermal fragmentation from ionic fragmentation with our measurement techniques estimating uncertainties is not feasible.

Line 317: Did the authors do similar test for organic nitrate products (-HNO3)?

**Answer:** Please see the answer to comment of previous reviewer regarding lines 307-309 in the manuscript.

Line 342: Should be "percentage".

**Answer:** Corrected.

Line 354: Should be "previous".

**Answer:** Corrected.

Figure 2: The error bar for the green dots need a darker color since it's hard to see. The ACM C*s don't have any error bar. Why, too small? In addition, since only averages were presented, it'd be worth to mention in section 2.4 that you calculated C* from equation 2 using how many samples for each technique for each experiment. As temperature varied though an experiment, how was C* affected?

**Answer:** Colors were changed for Figure 2. ACM had the lowest time resolution thus the number of data points was the lowest in comparison to the other instruments. This is the main reason why the error bars are lower for ACM in comparison to TD and CHARON. In Section 2.4 additional sentences were added:

*"Calculation of the average C* for every experiment was performed based on the time resolution of each instrument (section 2.2). When the signal in the particle-phase was close to the detection limit and introduced a high uncertainty, the calculation of the C* was not performed."*

Figure S6: Temp = 280K, why inconsistent with 298K used for experimental measurement mentioned at line 256? The name for each theoretical calculation is not consistent with that in the text, e.g., nano vs. NN.

**Answer:** The Figure S7 (see also answer to Reviewer 1) was added in order to be consistent with the temperature and the text. Also, this Figure was used in comparison to S6 to determine additional uncertainties in the theoretical calculations use during the subsequent analysis. For further details please see also answers to Reviewer 1 about uncertainty in theoretical vapor pressure calculation.

**References**

Gkatzelis, G. I., Tillmann, R., Hohaus, T., Müller, M., Eichler, P., Xu, K. M., Schlag, P., Schmitt, S. H., Wegener, R., Kaminski, M., Holzinger, R., Wisthaler, A., and Kiendler-Scharr, A.: Comparison of three aerosol chemical characterization techniques utilizing ptr-tof-ms: A study on freshly formed and aged biogenic soa, Atmos Meas Tech, 11, 1481-1500, 10.5194/amt-11-1481-2018, 2018.

[Figure]

**Figure S8: Characteristic example of the timeseries of C$_9$H$_{14}$O for the three different inlet techniques**